# On Gap-dependent Bounds for Offline Reinforcement Learning

**Xinqi Wang**
Institute for Interdisciplinary Information Sciences
Tsinghua University
wangxqkaxdd@gmail.com

**Qiwen Cui**
Paul G. Allen School of Computer Science
Engineering
University of Washington
qwcui@cs.washington.edu

**Simon S. Du**
Paul G. Allen School of Computer Science
Engineering
University of Washington
ssdu@cs.washington.edu

## Abstract

This paper presents a systematic study on gap-dependent sample complexity in offline reinforcement learning. Prior work showed when the density ratio between an optimal policy and the behavior policy is upper bounded (the optimal policy coverage assumption), then the agent can achieve an $O\left(\frac{1}{\epsilon^2}\right)$ rate, which is also minimax optimal. We show under the optimal policy coverage assumption, the rate can be improved to $O\left(\frac{1}{\epsilon}\right)$ when there is a positive sub-optimality gap in the optimal $Q$-function. Furthermore, we show when the visitation probabilities of the behavior policy are uniformly lower bounded for states where an optimal policy's visitation probabilities are positive (the uniform optimal policy coverage assumption), the sample complexity of identifying an optimal policy is independent of $\frac{1}{\epsilon}$. Lastly, we present nearly-matching lower bounds to complement our gap-dependent upper bounds.

## 1 Introduction

Reinforcement Learning (RL) aims to learn a policy that maximizes the long-term reward in unknown environments [Sutton and Barto, 2018]. The success of reinforcement learning often relies on being able to deploy the algorithms that directly interact with the environment. However, such direct interactions with real environments can be expensive or even impossible in many real-world applications, e.g., health and medicine [Murphy et al., 2001, Gottesman et al., 2019], education [Mandel et al., 2014], conversational AI [Ghandeharioun et al., 2019] and recommendation systems [Chen et al., 2019]. Instead, we have access to a dataset generated from some past suboptimal policies. Offline reinforcement learning (offline RL) aims to find a near-optimal policy using the offline dataset, and has achieved promising empirical successes [Lange et al., 2012, Levine et al., 2020].

Recently, a line of works showed that under the single policy coverage assumption (Assumption 3.2), one can obtain a near-optimal policy with polynomial number of samples [Rashidinejad et al., 2021, Xie et al., 2021b, Yin and Wang, 2021, Xie et al., 2021a, Li et al., 2022]. In particular, for the tabular setting, recent works have obtained minimax optimal sample complexity bounds $\widetilde{\Theta}\left(\frac{H^3 SC^*}{\epsilon^2}\right)$ where $H$ is the planning horizon, $S$ is the number of states, $C^*$ is the constant for the single policy coverage assumption, and $\epsilon$ is the target accuracy [Xie et al., 2021b, Li et al., 2022]. The $O\left(1/\epsilon^2\right)$ is for the

worst case and in many benign settings, one may use much fewer samples to learn a (near-)optimal policy.

How the benign problem structures help reduce the sample complexity has been extensively studied in the online bandits and reinforcement learning [Simchowitz and Jamieson, 2019, Jonsson et al., 2020, Wagenmaker et al., 2021b, Xu et al., 2021]. In particular, when there is a suboptimality gap between the optimal policy and the rest, then one can obtain $\log T$-type regret in contrast to $\sqrt{T}$-type regret bounds in the worst case where $T$ is the number of interactions. However, to our knowledge, how the gap structure helps reduce sample complexity in offline RL has not been thoroughly investigated. This paper presents a systematic study on gap-dependent bounds for offline RL in the canonical tabular setting, with nearly-matching upper and lower bounds in different regimes.

## 1.1 Main Contributions

We present novel analyses for the standard VI-LCB algorithm (Algorithm 2). Our main results are summarized in Table 1.1.

1. We develop a novel technique, *deficit thresholding*, to obtain gap-dependent bounds in offline RL. Different from the clip trick widely used in online RL [Simchowitz and Jamieson, 2019, Xu et al., 2021] to obtain gap-dependent bounds, our deficit thresholding technique is adaptive to the problem-instance. As will be shown in Section 6.2, this technique helps reduce the dependency on $H$ by adapting to the variance of the estimate.

2. Using the deficit thresholding technique, we obtain the first gap-dependent bound under the optimal policy coverage assumption. Specifically, we obtain an $\widetilde{O}\left(\frac{H^4 SC^*}{\epsilon \mathrm{gap}_{\min}}\right)$ bound where $\mathrm{gap}_{\min}$ is the minimum suboptimality gap between the optimal $Q$-value of the best action and that of the second-to-the-best action.[1] Notably, compared with the worst-case gap-indepdent, the rate improves from $1/\epsilon^2$ to $1/\epsilon$.

3. We also present the first gap-dependent lower bound for offline RL, $\Omega\left(\frac{H^2 SC^*}{\epsilon \mathrm{gap}_{\min}}\right)$ to show the $O(1/\epsilon)$ is unimprovable even with the gap condition, and our upper bound is tight up to an $H^2$ factor. The main insight from our lower bound is that if there exists a state whose visitation probability of the behavior policy and the optimal policy is $O(\epsilon)$, then we cannot learn much of the state using the offline dataset and will inevitably incur an $O(\epsilon)$ error.

4. We further study what condition permits even faster rate than $O(1/\epsilon)$. Leveraging our lower bound mentioned above, we propose a new condition, *uniform optimal policy coverage*, which posits that the visitation probabilities of the behavior policy are uniformly lower bounded by $P$ for states where an optimal policy's visitation probabilities are positive. Under this assumption, we obtain an $\widetilde{O}\left(\frac{H^3}{P \mathrm{gap}_{\min}^2}\right)$ bound. Importantly, this bound is *independent of* $1/\epsilon$ (not even $\log(1/\epsilon)$), a.k.a., one can identify an *exact optimal* policy. We also complement this upper bound with an $\Omega\left(\frac{H}{P \mathrm{gap}_{\min}^2}\right)$ lower bound to show our upper bound is tight up to an $H^2$ factor.

Lastly, we note that all of our bounds are obtained with the same algorithm, i.e., the algorithm automatically exploits the benign problem structure without any prior knowledge.

## 2 Related Work

We focus on existing theoretical results on gap-dependent bounds and offline RL in the tabular setting.

**Theoretical Results on Offline Tabular RL.** Theoretical analysis of offline RL can be traced back to Szepesvári and Munos [2005], under the uniform coverage assumption where *every state-action*

---

[1]The gap-dependent bounds in the online setting also depend on the suboptimality gaps of the actions other than the second-to-the-best action. This dependency is *not needed* for offline RL because even if the dataset does not have any information about the sub-optimal actions, the agent can still learn a near-optimal policy (as long as the dataset covers an optimal action). On the other hand, this dependency is needed in the online setting because the agent needs to explore the all actions.

| Condition | Upper bound | Lower bound |
|---|---|---|
| $C^*$ | $\widetilde{O}\left(\frac{H^3 S C^* \log\frac{1}{\delta}}{\epsilon^2}\right)$ | $\Omega\left(\frac{H^3 S C^*}{\epsilon^2}\right)$ |
| $C^*, \mathrm{gap_{min}}$ | $\widetilde{O}\left(\frac{H^4 S C^* \log\frac{1}{\delta}}{\epsilon \mathrm{gap_{min}}}\right)$ | $\Omega\left(\frac{H^2 S C^*}{\epsilon \mathrm{gap_{min}}}\right)$ |
| $P, \mathrm{gap_{min}}$ | $\widetilde{O}\left(\frac{H^3 \log\frac{1}{\delta}}{P \mathrm{gap_{min}^2}}\right)$ | $\Omega\left(\frac{H}{P \mathrm{gap_{min}^2}}\right)$ |

Table 1: Sample Complexity Bounds for different conditions about the sub-optimality and coverage. Cells in gray are the contributions of this work. The results in the first line without suboptimality gap assumptions were obtained in Xie et al. [2021b], Li et al. [2022]. $C^*$ stands for the relative optimal policy coverage coefficient, i.e., $\max\limits_{h,s}\frac{d_h^*(s)}{d_h^\mu(s)}$ where $d_h^*(s)$ is the visitation probability of the optimal policy for state $s$ at level $h$ and $d_h^\mu(s)$ is the visitation probability of the behavior policy $\mu$ for state $s$ at level $h$. $P$ stands for the uniform optimal policy coverage coefficient, i.e., $\min\limits_{h,s|d_h^*(s)>0} d_h^\mu(s)$. $\mathrm{gap_{min}}$ is the minimum non-zero suboptimality gap among all time-state-action tuples, i.e., $\min\limits_{h,s,a|a \text{ is not optimal}} \mathbf{V}_h^*(s) - \mathbf{Q}_h^*(s,a)$.

pairs are visted by the behavior policy with a positive probability. Sharp sample complexity bounds have been obtained under this assumption [Xie and Jiang, 2021, Xie et al., 2019, Yin et al., 2020, 2021b, Ren et al., 2021]. Recently, a line of works showed that under a much weaker assumption, single policy coverage, one can design sample efficient algorithms with both model-based and model-free methods based on the pessimism principle, [Rashidinejad et al., 2021, Yin and Wang, 2021, Xie and Jiang, 2021, Jin et al., 2021, Uehara and Sun, 2021, Uehara et al., 2021, Zanette et al., 2021]. Recently, Yin and Wang [2021] obtained a problem-dependent sample complexity in terms of the variance.

**Instance-dependent Sample Complexity in Online Learning and Generative Models.**    In online RL, a line of work studied the upper and lower bounds of gap-dependent sample complexity in both the regret and PAC settings [Simchowitz and Jamieson, 2019, Xu et al., 2021, Dann et al., 2021, Jonsson et al., 2020, Wagenmaker et al., 2021b,a, Tirinzoni et al., 2021]. Besides gap-dependent bounds, there are other problem-dependent bounds such as first-order and variance-dependent bounds Zanette and Brunskill [2019]. For generative models, Zanette et al. [2019] derived an upper bound depending on both variance and gap information.

**Gap-dependent bounds in Offline Learning.**    The most related work is by Hu et al. [2021] who studied the convergence rate of $Q$-learning in the discounted MDPs under the uniform coverage assumption.    They proved that one can obtain an $\epsilon$-optimal policy with $O\left(S^3 A^3 \log(1/\epsilon)/(1-\gamma)^4 P^2 \mathrm{gap_{min}^2}\right)$ samples for tabular MDP and $\widetilde{O}(1/\epsilon)$ (ignoring other parameters) for linear MDP. In comparison, we show that under the weaker uniform optimal policy coverage assumption, we can identify an exact optimal policy with $O\left(H^3/P\mathrm{gap_{min}^2}\right)$ sample complexity, which has no dependency on $1/\epsilon$.

## 3    Preliminaries

**Notations.**    We let $[n] = \{1, 2 \cdots n\}$. For two vectors $a, b$ of the same length $k$, we use $a \circ b$ to denote the Hadamard product $(a_1 b_1, a_2 b_2 \cdots a_k b_k)$. We use the standard definitions of $O(\cdot), \Theta(\cdot), \Omega(\cdot)$ to hide absolute constants, and tilded notations $\widetilde{O}(\cdot), \widetilde{\Theta}(\cdot), \widetilde{\Omega}(\cdot)$ to hide absolute constants as well as poly-logarithmic factors except for $\log\frac{1}{\delta}$. $\mathbf{Var}_p(V) = p^\top V \circ V - (p^\top V)^2$ refers to the variance of $V$ with respect to the weight $p$. $a \lesssim b$ means $a \leq Cb$ for some positive absolute constant $C$, and similarly $a \gtrsim b$ means $a \geq Cb$. $\mathbb{I}\{\xi\}$ is the indicator function of the event $\xi$, which equals 1 when $\xi$ is true and 0 otherwise. $\Delta(\mathcal{X})$ is the probability simplex over $\mathcal{X}$.

## 3.1 Markov Decision Processes

We consider tabular finite-horizon time-inhomogeneous MDPs described by the tuple $\mathcal{M} = (\mathcal{S}, \mathcal{A}, H, \mathcal{P}, p_0, r)$. Here $\mathcal{S}$ is the state space with cardinality $S$, $\mathcal{A}$ is the action space with cardinality $A$, $\mathcal{P} = \{p_{h,s,a}\}_{(h,s,a) \in [H] \times \mathcal{S} \times \mathcal{A}}$ with $p_{h,s,a} \in \Delta(\mathcal{S})$ is the transition kernel at timestep $h$, state $s$ and action $a$, $p_0 \in \Delta(\mathcal{S})$ is the initial state distribution of $s_1$, and $r = \{r_1, r_2, \ldots, r_H\}$ with $r_h : \mathcal{S} \times \mathcal{A} \to [0, 1]$ is the reward function.[2] For each episode, the player will generate a trajectory $\{(s_h, a_h, r_h)\}_{h=1}^H$ where $s_1 \sim p_0$, $s_{h+1} \sim p_{h,s_h,a_h}$, and $r_h = r_h(s_h, a_h)$, by controlling the actions $\{a_h\}_{h=1}^H$. The target of the player is to maximize the total reward $\sum_{h=1}^H r_h$.

**Policies.** A policy $\pi = \{\pi_h(\cdot|s)\}_{(h,s) \in [H] \times \mathcal{S}}$ refers to a set of distributions over $\mathcal{A}$. With a slight abuse of the notations, when a policy $\pi$ is deterministic, we use $\pi_h(s)$ to denote the action taken at timestep $h$ and state $s$. We define $\mathbb{E}_{\pi,\mathcal{M}}[\cdot] \triangleq \mathbb{E}_{\phi \sim (\pi,\mathcal{M})}[\cdot]$, where $\phi = \{(s_h, a_h, r_h)\}_{h \in [H]}$ is a trajectory sampled using policy $\pi$ in the MDP $\mathcal{M}$ and the expectation is over the randomness of both the policy and the transitions. $\mathcal{M}$ will be omitted when there is no confusion.

**Value Functions, Q-Functions and Policy Distributions.** For a given policy $\pi$ and an MDP $\mathcal{M}$, we define the state value function and the state-action value function to be

$$\mathbf{V}_h^\pi(s) \triangleq \mathbb{E}_\pi \left[ \sum_{h'=h}^H r_{h'}(s_{h'}, a_{h'}) \mid s_h = s \right], \mathbf{Q}_h^\pi(s, a) \triangleq \mathbb{E}_\pi \left[ \sum_{h'=h}^H r_{h'}(s_{h'}, a_{h'}) \mid s_h = s, a_h = a \right].$$

We define the optimal Q-function as $\mathbf{Q}_h^*(s, a) \triangleq \sup_\pi \mathbf{Q}_h^\pi(s, a)$, and similarly $\mathbf{V}_h^*(s) \triangleq \sup_\pi \mathbf{V}_h^\pi(s)$ for all $(h, s, a) \in [H] \times \mathcal{S} \times \mathcal{A}$. It is well known that there exists a deterministic optimal policy $\pi^*$ that can achieve the above maximum for all $s \in \mathcal{S}$, $a \in \mathcal{A}$ and $h \in [H]$ simultaneously.

We denote the value of a policy by $\mathbf{V}_0^\pi = \mathbb{E}_\pi \left[ \sum_{h=1}^H r_h(s_h, a_h) \right]$, and the value of the optimal policy by $\mathbf{V}_0^* = \mathbf{V}_0^{\pi^*}$. A policy $\pi$ is $\epsilon$-optimal if

$$\text{Suboptimal}(\pi) \triangleq \mathbf{V}_0^* - \mathbf{V}_0^\pi \le \epsilon.$$

We use $d_h^\pi(\cdot)$ to denote the probability of reaching a state $s$ under policy $\pi$:

$$d_h^\pi(s) \triangleq \mathbb{E}_\pi[\mathbb{I}\{s_h = s\}], \ d_h^\pi(s, a) \triangleq \mathbb{E}_\pi[\mathbb{I}\{(s_h, a_h) = (s, a)\}] = d_h^\pi(s)\pi_h(a \mid s).$$

**Sub-optimality Gap.** For a MDP instance $\mathcal{M}$, we define the gap at $(h, s, a) \in [H] \times \mathcal{S} \times \mathcal{A}$ to be $\text{gap}_h(s, a) = \mathbf{V}_h^*(s) - \mathbf{Q}_h^*(s, a)$, which is always non-negative because of the definition of $\pi^*$. Our results will depend on the smallest positive gap: $\text{gap}_{\min} \triangleq \min_{(h,s,a)|\text{gap}_h(s,a)>0} \text{gap}_h(s, a)$, which quantifies the difficulty of learning the optimal policy in the MDP instance.[3]

## 3.2 Offline Reinforcement Learning

**Offline Learning.** For offline reinforcement learning, we want to solve an MDP $\mathcal{M} = (\mathcal{S}, \mathcal{A}, H, \mathcal{P}, r)$ with unknown transitions by utilizing a given dataset collected by some unknown behavior policy $\mu$. Note that the algorithm is not allowed to perform any kind of additional sampling. The dataset is $\mathcal{D} = \{\{(s_{h,i}, a_{h,i}, r_{h,i})\}_{h \in [H]}\}_{i \in [N]}$, which contains $N$ trajectories collected by the behavior policy $\mu$ independently. A $(\epsilon, \delta)$-PAC offline reinforcement learning algorithm is defined to output an $\epsilon$-optimal policy $\pi$ with probability at least $1 - \delta$.

**Assumptions.** We introduce two dataset assumptions that will be used in this paper.

**Assumption 3.1** (Uniform optimal policy coverage). *We define the uniform optimal policy coverage coefficient to be*

$$P \triangleq \min_{\pi^*} \min_{h,s:d_h^{\pi^*}(s,a)>0} d_h^\mu(s, a),$$

*where $\pi^*$ is an optimal policy. We assume that $P > 0$.*

---

[2] We assume a known deterministic reward function as the main difficulty lies in learning the transition probability. All the conclusions in this paper can be proved for MDPs with unknown 1-subGuassian reward.

[3] We assume that at least one gap is positive. Otherwise all the actions are optimal and no learning is needed.

---
**Algorithm 1:** VI-LCB
---
    **input** : Dataset $\mathcal{D}_0$, reward function $r$

**1** Set $\underline{\mathbf{Q}}_{H+1}(s,a) = 0$

**2** Set $\underline{\mathbf{V}}_{H+1}(s,a) = 0$

**3** **for** $h \leftarrow H$ **to** $1$ **do**

**4**     Compute the empirical transition kernel $\hat{P}_h$

**5**     $\hat{P}_{h,s,a}(s') = \frac{N_h(s,a,s')}{N_h(s,a)}$ with $0/0 = 0$

**6**     **for** $s \in \mathcal{S}, a \in \mathcal{A}$ **do**

**7**         $b_h(s,a) \leftarrow C_b\sqrt{\frac{\mathbf{Var}_{\hat{P}_{h,s,a}}(\underline{\mathbf{V}}_{h+1})\iota}{N_h'(s,a)}} + C_b\frac{H\iota}{N_h'(s,a)}$, where $N_h'(s,a) = N_h(s,a) \vee \iota$

**8**         $\underline{\mathbf{Q}}_h(s,a) \leftarrow \max\{0, r_h(s,a) + \hat{P}_{h,s,a}^{\top}\underline{\mathbf{V}}_{h+1} - b_h(s,a)\}$

**9**     **for** $s \in \mathcal{S}$ **do**

**10**         $\underline{\mathbf{V}}_h(s) \leftarrow \max_{a \in \mathcal{A}} \underline{\mathbf{Q}}_h(s,a)$

**11**         $\underline{\pi}_h(s) \leftarrow \arg\max_{a \in \mathcal{A}} \underline{\mathbf{Q}}_h(s,a)$

    **output** : policy $\underline{\pi}$
---

---
**Algorithm 2:** Subsampled VI-LCB
---
    **input** : Dataset $\mathcal{D}$, reward function $r$

**1** Split $\mathcal{D}$ into 2 halves containing same number of sample trajectories, $\mathcal{D}^{\mathrm{main}}$ and $\mathcal{D}^{\mathrm{aux}}$

**2** $\mathcal{D}_0 = \{\}$

**3** **for** $(h,s) \in [H] \times \mathcal{S}$ **do**

**4**     $N_h^{\mathrm{trim}}(s) \leftarrow \max\{0, N_h^{\mathrm{aux}}(s) - 10\sqrt{N_h^{\mathrm{aux}}(s)\log\frac{HS}{\delta}}\}$

**5**     Randomly subsample $\min\{N_h^{\mathrm{trim}}(s), N_h^{\mathrm{main}}(s)\}$ samples of transition from $(h,s)$ from $\mathcal{D}^{\mathrm{main}}$ to add to $\mathcal{D}_0$

**6** $\underline{\pi} \leftarrow$ VI-LCB$(\mathcal{D}_0, r)$

    **output** : policy $\underline{\pi}$
---

Assumption 3.1 states that the behavior policy $\mu$ covers all the state-action pairs that some $\pi^*$ will choose with positive probability. This is a natural assumption if we want to recover the optimal policy.

A closely related assumption is the uniform coverage assumption, i.e., all $(h,s,a)$ tuples are covered by the dataset [Yin and Wang, 2020, Ren et al., 2021, Yin et al., 2021a, Hu et al., 2021]. Our assumption is significantly weaker as it only assumes covering the optimal policy. We will prove that under this assumption, we can identify the optimal policy with finite samples.

**Assumption 3.2** (Optimal policy coverage). *We define the relative optimal policy coverage coefficient to be*

$$C^* \triangleq \max_{\pi^*} \max_{(h,s,a) \in [H] \times \mathcal{S} \times \mathcal{A}} \frac{d_h^{\pi^*}(s,a)}{d_h^{\mu}(s,a)}$$

*with convention that $0/0 = 0$, where $\pi^*$ is an optimal policy. We assume that $C^* < \infty$.*

Similar coverage assumption has been widely adopted in Li et al. [2022], Shi et al. [2022], Yan et al. [2022], Jin et al. [2021], Rashidinejad et al. [2021]. Researchers have designed algorithms based on the pessimism principle to efficiently solve offline RL problems under this assumption. Assumption 3.2 is usually weaker than Assumption 3.1. For example, if there exists one unique optimal policy $\pi^*$ and $\mu = \pi^*$, we have $C^* = 1$ while $P$ can still be arbitrarily small. In addition, we always have $C^* \leq \frac{1}{P}$.

## 3.3 Subsampled VI-LCB

We briefly introduce the algorithm (Algorithm 2) that will be used in the analysis. Value Iteration with Lower Confidence Bound (VI-LCB) was first introduced by Rashidinejad et al. [2021] and improved by Li et al. [2022]. The main idea is to maintain a pessimistic estimate on the value functions so that the suboptimality of the output policy only depends on the uncertainty of the optimal policy. By utilizing the subsampling technique and Bernstein-style bonus, subsampled VI-LCB achieves the minimax sample complexity $\widetilde{O}(\frac{H^3 SC^*}{\epsilon^2})$.

In the algorithm, $N_h(s, a)$ refers to the number of sample transitions starting from state $s$, taking action $a$ at time step $h$ of some given dataset, and $N_h(s) = \sum_{a \in \mathcal{A}} N_h(s, a)$. Superscripts stand for the dataset. See Li et al. [2022] for a more detailed description of the algorithm.

## 4 Finding an Exact Optimal Policy with Assumption 3.1

In this section, we show that we can identify the exact best policy with finite samples by utilizing the gap structure under Assumption 3.1. On the other hand, for the minimax sample complexity $\widetilde{O}(H^3 SC^* \epsilon^{-2})$, it will become infinite when $\epsilon$ approaches 0. Note that directly setting $\epsilon < \text{gap}_{\min}$ does not imply that the output policy $\pi$ is optimal. Instead, we only have

$$\mathbf{V}_0^\pi \geq \mathbf{V}_0^* - \epsilon,$$

while $\pi$ can still be suboptimal at states visited with low probability.

**Theorem 4.1.** *For an MDP $\mathcal{M}$ and a behavior policy $\mu$ with uniform optimal coverage coefficient $P$, if the number of sample trajectories satisfies*

$$N \geq \widetilde{O}\left( \frac{H^3 \log \frac{1}{\delta}}{P\text{gap}_{\min}^2} \right),$$

*then with probability at least $1 - \delta$, Algorithm 2 returns an optimal policy.*

*Sketch of Proof.* First, if for all $(h, s, a) \in [H] \times \mathcal{S} \times \mathcal{A}$ satisfying $d_h^*(s, a) > 0$, we have

$$\underline{\mathbf{Q}}_h(s, a) > \mathbf{Q}_h^*(s, a) - \text{gap}_{\min}, \tag{1}$$

then we have $\underline{\pi}$ is an optimal policy. This is because as $\underline{\mathbf{Q}}$ is a pessimistic estimate of $Q^*$, we have

$$\underline{\mathbf{Q}}_h(s, a) > \mathbf{Q}_h^*(s, a) - \text{gap}_{\min} \geq \mathbf{Q}_h^*(s, a') \geq \underline{\mathbf{Q}}_h(s, a'),$$

for any action $a'$ that is sub-optimal. As a result, $\underline{\pi}$ chooses the optimal action for all $(h, s, a) \in [H] \times \mathcal{S} \times \mathcal{A}$ covered by the optimal policy, which implies $\underline{\pi}$ is an optimal policy.

Second, we show that (1) is satisfied with $N \geq \widetilde{O}\left( \frac{H^4}{P\text{gap}_{\min}^2} \right)$. This is because if the number of samples at $(h, s, a)$ exceeds $\widetilde{O}(H^4/\text{gap}_{\min}^2)$, we can guarantee that the estimation error at that step is smaller than $\text{gap}_{\min}/H$ by Hoeffding's inequality. Then the accumulated estimation error at $\underline{\mathbf{Q}}_h(s, a)$ can be bounded by $(H - h)\text{gap}_{\min}/H < \text{gap}_{\min}$, which implies (1). To further improve the dependence on $H$, we will use Bernstein's inequality and the proof is deferred to Appendix B.4. $\qquad\square$

## 5 Gap-dependent Upper Bounds with Assumption 3.2

In the previous section, we show that the optimal policy can be identified if Assumption 3.1 is satisfied. However, the uniform optimal coverage coefficient $P$ can be very small, which makes the bound $N \geq \widetilde{O}\left( \frac{H^3}{P\text{gap}_{\min}^2} \right)$ useless. In this section, we present two results on learning an $\epsilon$-optimal policy with assumption (Assumption 3.2) and we provide the proof sketch in the next section. The full proof can be found in Appendix B.

**Theorem 5.1.** *For an MDP $\mathcal{M}$ and behavior policy $\mu$ with relative optimal policy coverage coefficient $C^*$, if the number of sample trajectories satisfies*

$$N \geq \widetilde{O}\left( \frac{H^4 SC^* \log \frac{1}{\delta}}{\epsilon \text{gap}_{\min}} \right),$$

*Algorithm 2 returns an $\epsilon$-suboptimal policy with probability at least $1 - \delta$.*

Theorem 5.1 shows that to learn an $\epsilon$-optimal policy, we only need $\widetilde{O}(1/\epsilon\mathrm{gap}_{\min})$ samples, which significantly improves the minimax sample complexity $\widetilde{O}(1/\epsilon^2)$ when $\epsilon \ll \mathrm{gap}_{\min}$.

Now we show that we can further improve the bound if both Assumption 3.1 and Assumption 3.2 are satisfied.

**Theorem 5.2.** *For an MDP $\mathcal{M}$ and behavior policy $\mu$ with uniform optimal policy coverage coefficient $P$ and relative optimal policy coverage coefficient $C^*$, if the number of sample trajectories satisfies*

$$N \geq \widetilde{O}\left(\frac{H^3 S C^* \log\frac{1}{\delta}}{\epsilon\mathrm{gap}_{\min}} + \frac{H\log\frac{1}{\delta}}{P}\right),$$

*Algorithm 2 returns an $\epsilon$-suboptimal policy with probability at least $1 - \delta$.*

Theorem 5.2 improves an $H$ factor compared with Theorem 5.1, with the cost of an extra $H/P$ term. As the additional term is a constant with respect to $\epsilon$ and $\mathrm{gap}_{\min}$, the bound is improved when $\epsilon\mathrm{gap}_{\min}$ is small. We believe this $H/P$ can be removed and we leave it to future works.

# 6 Main Proof Techniques

In this section, we will provide a proof sketch for Theorem 5.1 and Theorem 5.2.

## 6.1 Pessimistic Algorithms

First, we define pessimistic algorithms and imaginary MDPs $\underline{\mathcal{M}} = (\mathcal{S}, \mathcal{A}, H, \mathcal{P}, \underline{r})$ determined by the pessimistic algorithms. Our analysis will generally hold for all pessimistic algorithms defined by Definition A.2 and we will show that VI-LCB (Algorithm 2) is a pessimistic algorithm. We use VI-LCB only to derive the final sample complexity guarantees.

**Definition 6.1** (Pessimistic algorithms). *An offline learning algorithm with output policy $\underline{\pi}$ is pessimistic if with probability at least $1 - \delta$, the following arguments hold,*

1. *It maintains a pessimistic estimate $\underline{\mathbf{Q}}$ of the true $Q^*$.*

2. *$\underline{\mathbf{Q}}$ is the optimal $Q$ function of an imaginary MDP $\underline{\mathcal{M}} = (\mathcal{S}, \mathcal{A}, H, \mathcal{P}, p_0, \underline{r})$, where $\underline{r}_h(s,a) \leq r_h(s,a)$ for all $(h,s,a) \in [H] \times \mathcal{S} \times \mathcal{A}$.[4]*

3. *$\underline{\pi}$ is the greedy policy with respect to $\underline{\mathbf{Q}}$.*

Most of the existing offline RL algorithms are pessimistic algorithms and we will also prove it for VI-LCB [Li et al., 2022] in Appendix B.

**Definition 6.2** (Deficit). *For any pessimistic algorithm and the corresponding imaginary MDP $\underline{\mathcal{M}} = (\mathcal{S}, \mathcal{A}, H, \mathcal{P}, p_0, \underline{r})$, we define the deficit to be*

$$\mathbf{E}_h(s,a) = r_h(s,a) - \underline{r}_h(s,a), \ \forall(h,s,a) \in [H] \times \mathcal{S} \times \mathcal{A}.$$

From the definition of pessimistic algorithms, we have $\mathbf{E}_h(s,a) \geq 0$ immediately. Intuitively, deficit stands for how pessimistic the estimates are. Note that the deficit is related to the algorithm itself and usually we can bound it by utilizing concentration inequalities.

## 6.2 Deficit Thresholding for Analysising LCB-style Algorithms

By defining $\underline{\mathbf{V}}^\pi$ to be the value function of policy $\pi$ in $\underline{\mathcal{M}}$ and recalling that $\underline{\pi}$ is the optimal policy in $\underline{\mathcal{M}}$, we have $\underline{\mathbf{V}}_0^* \leq \underline{\mathbf{V}}_0^{\underline{\pi}} \leq \mathbf{V}_0^{\underline{\pi}} \leq \mathbf{V}_0^*$ and thus

$$\mathbf{V}_0^* - \underline{\mathbf{V}}_0^* = \sum_{h=1}^{H} \mathbb{E}_{\pi^*}[r_h(s_h, a_h) - \underline{r}_h(s_h, a_h)] = \sum_{h=1}^{H} \mathbb{E}_{\pi^*}[\mathbf{E}_h(s_h, a_h)]$$

---

[4]Here we loosen the definition of MDP by allowing the reward function to have negative value. $\underline{r}$ maybe negative, but as will be shown in appendix, $\underline{\mathbf{V}}^{\underline{\pi}}$ and $\underline{\mathbf{Q}}^{\underline{\pi}}$ are still non-negative, thus does not affect our analysis.

can upper bound the suboptimality of $\underline{\pi}$. Surprisingly, we will show that even if we threshold the deficit, a similar upper bound still holds. Define the thresholded deficits and the corresponding reward functions as

$$\ddot{\mathbf{E}}_h(s,a) \triangleq \max\{0, \mathbf{E}_h(s,a) - \epsilon_h(s,a)\}, \; \ddot{r}_h(s,a) \triangleq r_h(s,a) - \ddot{\mathbf{E}}_h(s,a), \; \forall (h,s,a) \in [H] \times \mathcal{S} \times \mathcal{A},$$

for any non-negative threshold function $\epsilon_h(s,a)$. Then we can define a thresholded MDP $\ddot{\mathcal{M}} = (\mathcal{S}, \mathcal{A}, H, \mathcal{P}, \ddot{r})$ and we use $\ddot{\mathbf{V}}$ to denote the value function in $\ddot{\mathcal{M}}$. Now we present the key lemma showing that if the thresholding function satisfies certain conditions related to $\text{gap}_{\min}$, then the suboptimality of $\underline{\pi}$ can still be bounded by the thresholded deficit.

**Lemma 6.1.** *Suppose for a thresholding function $\epsilon_h(s,a)$, for all $(h,s) \in [H] \times \mathcal{S}$, we have*

$$\underline{\mathbf{V}}_h^*(s) + \frac{\text{gap}_{\min}}{2} \geq \ddot{\mathbf{V}}_h^*(s). \tag{2}$$

*Then we can bound the suboptimality of $\underline{\pi}$ by the thresholded deficit:*

$$\text{suboptimal}(\underline{\pi}) = \mathbf{V}_0^* - \mathbf{V}_0^{\underline{\pi}} \leq \mathbf{V}_0^* - \underline{\mathbf{V}}_0^* \leq 2(\mathbf{V}_0^* - \ddot{\mathbf{V}}_0^*) = 2\sum_{h=1}^{H} \mathbb{E}_{\pi^*}[\ddot{\mathbf{E}}_h(s_h, a_h)].$$

See Appendix A.2 for the rigorous proof, where this lemma is decomposed and restated as Theorem A.1 and Definition A.5. One way to satisfy (2) is to set $\epsilon_h(s,a) = \frac{\text{gap}_{\min}}{2H}$ for all $(h,s,a) \in [H] \times \mathcal{S} \times \mathcal{A}$. As a result, the reward at each timestep $h$ is increased by at most $\frac{\text{gap}_{\min}}{2H}$ after thresholding, so the overall increase of the value function is bounded by $\text{gap}_{\min}/2$. This kind of $\epsilon_h(s,a)$ leads to the following corollary.

**Corollary 6.1.** *For any pessimistic algorithm, we have*

$$\mathbf{V}_0^* - \mathbf{V}_0^{\underline{\pi}} \leq 2\sum_{h=1}^{H} \mathbb{E}_{\pi^*}\left[\max\{0, \mathbf{E}_h(s_h, a_h) - \frac{\text{gap}_{\min}}{2H}\}\right].$$

In addition, note that $\mathbf{E}_h(s_h, a_h)$ depends on variance of the transition $p_{h,s_h,a_h}$ as if the transition variance is small, the estimate will be accurate and the deficit will be small. This inspires us to threshold adaptively based on the variance. We design the following adaptive threshold function:

$$\epsilon_h(s,a) \propto \frac{\mathbf{Var}_{\hat{P}_{h,s,a}}(\mathbf{V}_{h+1})}{H^2} \text{gap}_{\min}.$$

It turns out that $\widetilde{\Omega}(\frac{H}{P})$ sample complexity is enough to guarantee this kind of threshold function satisfies (2). (See Lemma B.8 in Appendix B.5.1)

**Corollary 6.2.** *For any pessimistic algorithm and number of samples $N \geq \widetilde{O}(\frac{H}{P})$, we have*

$$\mathbf{V}_0^* - \mathbf{V}_0^{\underline{\pi}} \leq 2\sum_{h=1}^{H} \mathbb{E}_{\pi^*}\left[\max\{0, \mathbf{E}_h(s_h, a_h) - \frac{\mathbf{Var}_{\hat{P}_{h,s,a}}(\mathbf{V}_{h+1}^{\underline{\pi}})}{H^2} \text{gap}_{\min} - \frac{\text{gap}_{\min}}{4H}\}\right].$$

To proceed from Lemma 6.1, we will utilize the following inequality

$$\ddot{\mathbf{E}}_h(s,a) = \max\{0, \mathbf{E}_h(s,a) - \epsilon_h(s,a)\} \leq \frac{\mathbf{E}_h^2(s,a)}{\epsilon_h(s,a)}.$$

Then with different choices of $\epsilon_h(s,a)$ used in Corollary 6.1 and Corollary 6.2 and the bonus function used in the VI-LCB algorithm, we can derive Theorem 5.1 and Theorem 5.2. Here we briefly explain the $\widetilde{O}(1/\epsilon)$ dependence. We have $\mathbf{E}_h(s,a) \leq O(b_h(s,a))$, where $b_h(s,a)$ is the pessimistic bonus scaling as $O(1/\sqrt{n})$ (ignoring other dependences). Then by Lemma 6.1 and the above inequality, we can achieve an $\widetilde{O}(1/\epsilon)$ bound immediately. A more detailed proof is provided in Appendix B.

Our technique is similar to the clip techique [Simchowitz and Jamieson, 2019] in online MDP as both of them threshold the estimates (deficit here and surplus in their work) by some $\Theta(\text{gap}_{\min})$ terms. Simchowitz and Jamieson [2019] further clip another $\frac{\text{gap}_h(s,\bar{a})}{H}$ term, and we can actually achieve that as well, but it does not improve the sample complexity in the offline setting as we do not need to explore all the actions. In addition, we develop a new thresholding function based on the empirical variance, which can improve the $H$ factor by utilizing the Bernstein's inequality and the total variance technique. We believe this new technique can also be applied to the online setting and improve the sample complexity there.

# 7 Gap-dependent Lower Bounds

In this section, we provide two lower bounds for uniform optimal coverage assumption (Assumption 3.1) and optimal policy coverage assumption (Assumption 3.2) respectively. Our lower bounds show that Theorem 4.1 and Theorem 5.1 are optimal up to $H$ factors and logarithm terms. We begin with a general lower bound. Here the *offline learning algorithm*, **ALG**, is defined as the algorithm that takes a dataset $\mathcal{D}$ as input and output a policy $\hat{\pi}$. Note that **ALG** can be stochastic.

**Theorem 7.1.** *There exists some absolute constant $C$, such that for any $A \geq 3, S \geq 2, H \geq 2, \tau < \frac{1}{2}, \lambda \leq \frac{1}{3}, \lambda_1 \geq 2$ and offline RL algorithm **ALG**, if the number of sample trajectories satisfies*

$$N \leq C \cdot \frac{HS\lambda_1}{\lambda\tau^2},$$

*there exists a MDP instance $\mathcal{M}$ and a behavior policy $\mu$ with $\mathrm{gap}_{\min} = \tau$, $P \geq \frac{\lambda}{eS\lambda_1}$, $C^* \leq \lambda_1$ such that the output policy $\hat{\pi}$ of **ALG** has expected suboptimality*

$$\mathbb{E}_{\mathcal{M},\mu,\mathbf{ALG}}[\mathbf{V}_0^* - \mathbf{V}_0^{\hat{\pi}}] \geq \frac{\lambda H\tau}{12}.$$

By choosing $\lambda = 1/3$, $\lambda_1 = \frac{1}{3ePS}$ and $\tau = \mathrm{gap}_{\min}$, Theorem 7.1 indicates the following corollary.

**Corollary 7.1.** *For any given instance coefficients $(H, \mathrm{gap}_{\min}, P \leq \frac{1}{6eS})$, target suboptimality $\epsilon \lesssim H\mathrm{gap}_{\min}$ and any offline reinforcement learning algorithm **ALG**, there exists an instance $(\mathcal{M}, \mu)$ such that if the number of trajectory samples satisfies*

$$N \leq C \cdot \frac{H}{P\mathrm{gap}_{\min}^2},$$

*the output policy $\hat{\pi}$ would have expected suboptimality more than $\epsilon$, i.e.,*

$$\mathbb{E}_{\mathcal{M},\mu,\mathbf{ALG}}[\mathbf{V}_0^* - \mathbf{V}_0^{\pi^*}] \geq \frac{1}{36} H\mathrm{gap}_{\min} \geq \epsilon$$

By choosing $\lambda = \frac{12\epsilon}{H\mathrm{gap}_{\min}}$, $\lambda_1 = C^*$, $\tau = \mathrm{gap}_{\min}$, Theorem 7.1 indicates the following corollary.

**Corollary 7.2.** *For any given instance coefficients $(H, S, \mathrm{gap}_{\min}, C^* \geq 2)$, target suboptimality $\epsilon \lesssim H\mathrm{gap}_{\min}$ and any offline learning algorithm **ALG**, there exists an instance $(\mathcal{M}, \mu)$ such that if the number of trajectory samples satisfies*

$$N \leq C \cdot \frac{H^2 S C^*}{\epsilon\mathrm{gap}_{\min}},$$

*the output policy $\hat{\pi}$ would have expected suboptimality more than $\epsilon$, i.e.,*

$$\mathbb{E}_{\mathcal{M},\mu,\mathbf{ALG}}[\mathbf{V}_0^* - \mathbf{V}_0^{\pi^*}] \geq \epsilon.$$

The proof of the corollaries can be found in Appendix C.1. We construct a family of MDPs that involve solving $HS$ independent bandit problems, where each bandit requires $\widetilde{\Omega}(\frac{HA}{\tau^2})$ visits so that the estimation error can be bounded by $\tau$. Similar constructions have been made in Yin et al. [2021a], Dann et al. [2017], Xie et al. [2021b] . Our key observation is that the initial state distribution can linearly determine the uniform optimal policy coverage coefficient $P$ and the final suboptimality. Also, our proof can lead to another version of lower bound, where a $\Omega(\epsilon)$ suboptimality occurs with a constant probability. See Appendix C.2.3 for details.

# 8 Necessity of All-Optimal-Policy Coverage

As one may notice that Assumption 3.1 and Assumption 3.2 are not exactly the most widely adopted versions [Li et al., 2022, Xie et al., 2021b, Rashidinejad et al., 2021]. Here the assumptions require a coverage of all the optimal policies instead of only one optimal policy. This assumption can not be weaken as long as we are using naive LCB-style algorithm, and a counterexample can be found in Appendix D.

Intuitively, the problem lies in that we can not distinguish between different optimal policies. Gap information helps us distinguish optimal actions from sub-optimal ones when the estimation error is relatively small, but a naive LCB algorithm still has the probability of choosing an optimal action $((s, a_2)$ in the counterexample) without fully exploring the subsequent state-action pairs. In online gap-dependent analysis, this problem can be addressed by some sophisticated exploration techniques [Xu et al., 2021]. It is unclear if we can weaken the assumption to single policy coverage and we leave this to future work.

## 9 Conclusion

We presented a systematic study on gap-dependent upper and lower bounds for offline reinforcement learning. Depending on different assumptions, the rates can be improved from $\widetilde{O}(1/\epsilon^2)$ in the worst to $\widetilde{O}(1/\epsilon)$ or even independent of $1/\epsilon$. Also, we presented a counterexample to show that overall optimal policy coverage is necessary in gap-dependent analysis.

One open question is that there still exists a gap between the upper and lower bounds in terms of $H$. We note that this gap also exists in the online setting [Simchowitz and Jamieson, 2019, Xu et al., 2021]. Another direction is to generalize our results to the function approximation setting [He et al., 2021].

## Acknowledgements

SSD acknowledges funding from NSF Award's CCF-2212261, IIS-2143493, DMS-2134106, CCF-2019844 and IIS-2110170.

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
