# A Upper Bound with Gap-dependent Analysis

We begin with the proof of thresholding technique. In this section, definitions are restated for completeness.

## A.1 Definitions

We first restate the notations.

**Definition A.1** (Pessimistic algorithms). *An offline learning algorithm with output policy $\underline{\pi}$ is pessimistic if with probability at least $1 - \delta$, the following arguments hold,*

1. *It maintains a pessimistic estimate $\underline{\mathbf{Q}}$ of the true $Q^*$.*

2. *$\underline{\mathbf{Q}}$ is the optimal Q function of an imaginary MDP $\underline{\mathcal{M}} = (\mathcal{S}, \mathcal{A}, H, \mathcal{P}, p_0, \underline{r})$, where $\underline{r}_h(s, a) \leq r_h(s, a)$ for all $(h, s, a) \in [H] \times \mathcal{S} \times \mathcal{A}$.*

3. *$\underline{\pi}$ is the greedy policy with respect to $\underline{\mathbf{Q}}$.*

**Definition A.2** (Pessimistically estimated MDP). *For a given successful pessimistic algorithm execution instance, where the arguments in Definition A.1 are simultaneously satisfied, we call $\underline{\mathcal{M}} = (\mathcal{S}, \mathcal{A}, H, \mathcal{P}, p_0, \underline{r})$ the pessimistically estimated MDP. At the same time, $\underline{\mathbf{V}}$, $\underline{\mathbf{Q}}$ are the corresponding value functions and Q functions. We use $\underline{\pi}$ to refer to the returned policy, which is optimal over $\underline{\mathcal{M}}$.*

And sometimes we use $\underline{\mathbf{Q}} = \underline{\mathbf{Q}}^{\underline{\pi}}$, $\underline{\mathbf{V}} = \underline{\mathbf{V}}^{\underline{\pi}}$ without superscript indicating the policy. We'll show that this notation matches the definition in Algorithm 1 so there is no need worrying about any possible confusion.

**Definition A.3** (Deficit). *For a pessimistically estimated MDP $\underline{\mathcal{M}} = (\mathcal{S}, \mathcal{A}, H, \mathcal{P}, p_0, \underline{r})$, we define deficit to be*

$$\mathbf{E}_h(s, a) \triangleq r_h(s, a) - \underline{r}_h(s, a).$$

**Definition A.4** (Not-so-pessimistic MDP). *For a given set of $\epsilon_h(s, a)$, a pessimistically estimated MDP $\underline{\mathcal{M}} = (\mathcal{S}, \mathcal{A}, H, \mathcal{P}, p_0, \underline{r})$, define*

$$\ddot{r}_h(s, a) \triangleq r_h(s, a) - \max\{0, \mathbf{E}_h(s, a) - \epsilon_h(s, a)\}.$$

*Then we call $\ddot{\mathcal{M}} = (\mathcal{S}, \mathcal{A}, H, \mathcal{P}, p_0, \ddot{r})$ not-so-pessimistic MDP. At the same time, $\ddot{\mathbf{V}}$ is the corresponding value functions.*

## A.2 Main Theorem

For conciseness, we will use $a^*$ and $\underline{a}$ to stand for $\pi_h^*(s)$ and $\underline{\pi}_h(s)$ respectively when it introduces no confusion. To formally present the deficit thresholding technique, we define a event $\xi_{gap}$.

**Definition A.5** (Gap restriction event). *For a given set of $\epsilon_h(s, a)$, event $\xi_{gap}$ is defined to be the event such that for all optimal policy $\pi^*$, $h \in [H]$ and $s \in \mathcal{S}$,*

$$\ddot{\mathbf{V}}_h^*(s) \leq \underline{\mathbf{V}}_h^*(s) + \frac{\mathrm{gap}_{\min}}{2}.$$

Note that $\xi_{gap}$ depends on the value of $\epsilon_h(s, a)$, and the definition of $\epsilon_h(s, a)$ may involve randomness. In the following proof of Corollary A.1, we will set $\epsilon_h(s, a) = \frac{\mathrm{gap}_{\min}}{2H}$.

**Theorem A.1** (Deficit thresholding). *When event $\xi_{gap}$ happens, there exists an optimal policy $\pi^*$, such that replacing $\mathbf{V}_0^{\pi}$ with $\ddot{\mathbf{V}}_0^*$ only harms the difference up to a constant factor,*

$$\mathbf{V}_0^* - \mathbf{V}_0^{\pi} \leq 2(\mathbf{V}_0^* - \ddot{\mathbf{V}}_0^*).$$

Rigorous proof is deferred to Appendix A.4.

**Corollary A.1.** *For a pessimistic algorithm running instance, there exists a deterministic optimal policy $\pi^*$, such that*

$$\mathbf{V}_0^* - \mathbf{V}_0^{\pi} \leq 2 \sum_{h=1}^{H} \mathbb{E}_{\pi^*} \left[ \max\{0, \mathbf{E}_h(s_h, a_h) - \frac{\mathrm{gap}_{\min}}{2H}\} \right].$$

*Proof.* With Theorem A.1, we just need to prove that $\epsilon_h(s,a) = \frac{\text{gap}_{\min}}{2H}$ indicates $\xi_{gap}$.

Note that $\mathbf{E}_h(s,a^*) \leq \ddot{\mathbf{E}}_h(s,a^*) + \epsilon_h(s,a^*) = \ddot{\mathbf{E}}_h(s,a^*) + \frac{\text{gap}_{\min}}{2H}$. Therefore, for all optimal policy $\pi^*$, we have

$$\ddot{\mathbf{V}}_h^*(s) - \underline{\mathbf{V}}_h^*(s) = \sum_{h'=h}^{H} \mathbb{E}_{\pi^*, s_h=s}[-\ddot{\mathbf{E}}_{h'}(s_{h'}, a_{h'}) + \mathbf{E}_{h'}(s_{h'}, a_{h'})]$$

$$\leq \sum_{h'=h}^{H} \mathbb{E}_{\pi^*, s_h=s}[\frac{\text{gap}_{\min}}{2H}]$$

$$\leq \frac{\text{gap}_{\min}}{2}.$$

$\square$

## A.3 Value/Q Function Ranking Lemma

The following lemmas will be frequently used throughout the proof of Theorem A.1 and upper bounds.

**Lemma A.1** (Overall size relationships of value functions). *When $\xi_{gap}$ happens, different value functions satisfy that for any optimal policy $\pi^*$, we have*

$$\mathbf{V}^* \geq \begin{cases} \mathbf{V}^{\overline{\pi}} \geq \underline{\mathbf{V}}^{\overline{\pi}} \\ \ddot{\mathbf{V}}^* \end{cases} \geq \underline{\mathbf{V}}^* \geq \ddot{\mathbf{V}}^* - \frac{\text{gap}_{\min}}{2}.$$

*Here $\mathbf{V} \geq \mathbf{V}'$ means $\mathbf{V}_h(s) \geq \mathbf{V}'_h(s)$ for all $(h,s) \in [H] \times \mathcal{S}$.*

*Proof.* We study each inequality one by one:

$\mathbf{V}^* \geq \mathbf{V}^{\overline{\pi}}$: $\pi^*$ is a optimal policy over $\mathcal{M}$.

$\mathbf{V}^{\overline{\pi}} \geq \underline{\mathbf{V}}^{\overline{\pi}}$: this follows from $\underline{r} \leq r$,

$$\mathbf{V}_h^{\overline{\pi}}(s) = \sum_{h'=h}^{H} \mathbb{E}_{\overline{\pi}}[r_{h'}(s_{h'}, a_{h'}) \mid s_h = s] \geq \sum_{h'=h}^{H} \mathbb{E}_{\overline{\pi}}[\underline{r}_{h'}(s_{h'}, a_{h'}) \mid s_h = s] = \underline{\mathbf{V}}_h^{\overline{\pi}}(s).$$

$\underline{\mathbf{V}}^{\overline{\pi}} \geq \underline{\mathbf{V}}^*$: $\underline{\pi}$ is a optimal policy over $\underline{\mathcal{M}}$.

$\mathbf{V}^* \geq \ddot{\mathbf{V}}^*$: this follows from $\ddot{r} \leq r$,

$$\mathbf{V}_h^*(s) = \sum_{h'=h}^{H} \mathbb{E}_{\pi^*}[r_{h'}(s_{h'}, a_{h'}) \mid s_h = s] \geq \sum_{h'=h}^{H} \mathbb{E}_{\pi^*}[\ddot{r}_{h'}(s_{h'}, a_{h'}) \mid s_h = s] = \ddot{\mathbf{V}}_h^*(s).$$

$\ddot{\mathbf{V}}^* \geq \underline{\mathbf{V}}^*$: this follows from $\underline{r} \leq \ddot{r}$,

$$\ddot{\mathbf{V}}_h^*(s) = \sum_{h'=h}^{H} \mathbb{E}_{\pi^*}[\ddot{r}_{h'}(s_{h'}, a_{h'}) \mid s_h = s] \geq \sum_{h'=h}^{H} \mathbb{E}_{\pi^*}[\underline{r}_{h'}(s_{h'}, a_{h'}) \mid s_h = s] = \underline{\mathbf{V}}_h^*(s).$$

$\underline{\mathbf{V}}^* \geq \ddot{\mathbf{V}}^* - \frac{\text{gap}_{\min}}{2}$: this is just the definition of $\xi_{gap}$. $\square$

**Lemma A.2** (Overall size relationships of Q functions). *When $\xi_{gap}$ happens, different Q functions satisfy that for any optimal policy $\pi^*$, we have*

$$\mathbf{Q}_h^*(s,a^*) \geq \mathbf{Q}_h^*(s,\underline{a}) \geq \mathbf{Q}_h^{\overline{\pi}}(s,\underline{a}) \geq \underline{\mathbf{Q}}_h^{\overline{\pi}}(s,\underline{a}) \geq \underline{\mathbf{Q}}_h^{\overline{\pi}}(s,a^*) \geq \underline{\mathbf{Q}}_h^*(s,a^*).$$

*Proof.* We study each inequality one by one:

$\mathbf{Q}_h^*(s,a^*) \geq \mathbf{Q}_h^*(s,\underline{a})$: $a^*$ is the optimal action at $(h,s)$ over $\mathcal{M}$.

$\mathbf{Q}_h^*(s,\underline{a}) \geq \mathbf{Q}_h^{\overline{\pi}}(s,\underline{a})$: this follows from $\mathbf{V}^* \geq \mathbf{V}^{\overline{\pi}}$ in Lemma A.1,

$$\mathbf{Q}_h^*(s,\underline{a}) = \mathbb{E}_{s' \sim P_{h,s,\underline{a}}}[\mathbf{V}_{h+1}^*(s')] \geq \mathbb{E}_{s' \sim P_{h,s,\underline{a}}}[\mathbf{V}_{h+1}^{\overline{\pi}}(s')] = \mathbf{Q}_h^{\overline{\pi}}(s,\underline{a}).$$

$\mathbf{Q}_h^{\overline{\pi}}(s,\underline{a}) \geq \underline{\mathbf{Q}}_h^{\overline{\pi}}(s,\underline{a})$: this follows from $\mathbf{V}^{\overline{\pi}} \geq \underline{\mathbf{V}}^{\overline{\pi}}$ in Lemma A.1.

$\underline{\mathbf{Q}}_h^{\overline{\pi}}(s,\underline{a}) \geq \underline{\mathbf{Q}}_h^{\overline{\pi}}(s,a^*)$: $\underline{a}$ is the optimal action at $(h,s)$ over $\underline{\mathcal{M}}$.

$\underline{\mathbf{Q}}_h^{\overline{\pi}}(s,a^*) \geq \underline{\mathbf{Q}}_h^*(s,a^*)$: this follows from $\underline{\mathbf{V}}^{\overline{\pi}} \geq \underline{\mathbf{V}}^*$ in Lemma A.1,

$$\underline{\mathbf{Q}}_h^{\overline{\pi}}(s,a^*) = \mathbb{E}_{s' \sim P_{h,s,a^*}}[\underline{\mathbf{V}}_{h+1}^{\overline{\pi}}(s')] \geq \mathbb{E}_{s' \sim P_{h,s,a^*}}[\underline{\mathbf{V}}_{h+1}^*(s')] = \underline{\mathbf{Q}}_h^*(s,a^*).$$

$\square$

## A.4 Proof of Theorem A.1

*Proof.* In this proof, we choose the $\pi^*$ according to the given $\underline{\pi}$,

$$\pi_h^*(s) = \begin{cases} \underline{\pi}_h(s) & \underline{\pi}_h(s) \text{ is optimal,} \\ \text{arbitrary optimal action} & \underline{\pi}_h(s) \text{ is not optimal.} \end{cases}$$

So that every time $\pi^*$ disagrees with $\underline{\pi}$, the choice made by $\underline{\pi}$ must be suboptimal. The intuition is that we only consider the cases where $\pi^*$ and $\underline{\pi}$ have different opinions. To begin with, we define a set of prefix trajactories for any two given deterministic policies over MDPs that only differ in rewards:

$$\Psi(\pi_1, \pi_2) = \{(s_1, a_1, \cdots, s_k) \mid \pi_{1,i}(s_i) = \pi_{2,i}(s_i) = a_i, \forall i = 1, 2, \cdots k - 1,$$
$$\pi_1(s_k) \neq \pi_2(s_k) \text{ or } k = H\}.$$

And we use $P_\psi^\pi$ to denote the probability that we can get a prefix trajactory $\psi = (s_1, a_1, \cdots, s_k)$ with a deterministic policy $\pi$,

$$P_\psi^\pi \triangleq p_0(s_1) \prod_{h=1}^{k-1} p_h(s_h, \pi_h(s_h), s_{h+1}).$$

Notice that for any given trajectory $\xi$ and policy $\pi_1, \pi_2$, there is exactly one prefix trajectory $\psi \in \Psi(\pi_1, \pi_2)$ being the prefix of $\xi$, which ends at the first time $\pi_1$ disagrees with $\pi_2$. Denote the length and the last state of a trajectory to be $2h_\psi - 1$ and $s_\psi$, and set the cumulative reward over $\psi$ under a given deterministic reward function $r_h(s, a)$ to be $r_\psi$, we can write the value function in the form

$$V_0^{\pi_1} = \sum_{\psi \in \Psi(\pi_1, \pi_2)} P_\psi^{\pi_1} \left( r_\psi + V_{h_\psi}^{\pi_1}(s_\psi) \right).$$

Also, notice that because $\pi_1$ and $\pi_2$ agrees on all the decisions in $\psi \in \Psi(\pi_1, \pi_2)$, we always have $P_\psi^{\pi_1} = P_\psi^{\pi_2}$ for $\psi \in \Psi(\pi_1, \pi_2)$. Now we have

$$\mathbf{V}_0^* - \mathbf{V}_0^{\underline{\pi}} = \sum_{\psi \in \Psi(\pi^*, \underline{\pi})} P_\psi^{\pi^*} (\mathbf{V}_{h_\psi}^*(s_\psi) - \mathbf{V}_{h_\psi}^{\underline{\pi}}(s_\psi))$$

$$= \sum_{\psi \in \Psi(\pi^*, \underline{\pi})} P_\psi^{\pi^*} (\mathbf{Q}_{h_\psi}^*(s_\psi, \underline{a}) + \text{gap}_{h_\psi}(s_\psi, \underline{a}) - \mathbf{V}_{h_\psi}^{\underline{\pi}}(s_\psi)).$$

Then we prove a statement that $\forall \psi \in \Psi(\pi^*, \underline{\pi})$,

$$\mathbf{Q}_{h_\psi}^*(s_\psi, \underline{a}) + \text{gap}_{h_\psi}(s_\psi, \underline{a}) - \ddot{\mathbf{V}}_{h_\psi}^*(s_\psi) \geq \mathbf{Q}_{h_\psi}^*(s_\psi, \underline{a}) + \frac{1}{2}\text{gap}_{h_\psi}(s_\psi, \underline{a}) - \mathbf{V}_{h_\psi}^{\underline{\pi}}(s_\psi). \quad (3)$$

When $\underline{a} = a^*$, the only possibility is that $h_\psi = H$, then RHS=0. While the LHS is always non-negative because $\mathbf{Q}_{h_\psi}^*(s_\psi, \underline{a}) = \mathbf{V}_{h_\psi}^*(s_\psi) \geq \ddot{\mathbf{V}}_{h_\psi}^*(s_\psi)$ (Lemma A.1).
When $\underline{a} \neq a^*$, event $\xi_{gap}$ guarantees that

$$\mathbf{Q}_{h_\psi}^*(s_\psi, \underline{a}) + \text{gap}_{h_\psi}(s_\psi, \underline{a}) - \ddot{\mathbf{V}}_{h_\psi}^*(s_\psi) \geq \mathbf{Q}_{h_\psi}^*(s_\psi, \underline{a}) + \text{gap}_{h_\psi}(s_\psi, \underline{a}) - \underline{\mathbf{V}}_{h_\psi}^*(s_\psi) - \frac{\text{gap}_{\min}}{2}$$

$$\geq \mathbf{Q}_{h_\psi}^*(s_\psi, \underline{a}) + \frac{1}{2}\text{gap}_{h_\psi}(s_\psi, \underline{a}) - \mathbf{V}_{h_\psi}^{\underline{\pi}}(s_\psi).$$

The inequality uses that $\text{gap}_{\min} \leq \text{gap}_{h_\psi}(s_\psi, \underline{a})$ and that $\underline{\mathbf{V}}^* \leq \underline{\mathbf{V}}^{\underline{\pi}} \leq \mathbf{V}^{\underline{\pi}}$. At the same time, we can decompose $\mathbf{V}_0^* - \ddot{\mathbf{V}}_0^*$ in a similar way,

$$\mathbf{V}_0^* - \ddot{\mathbf{V}}_0^* = \sum_{\psi \in \Psi(\pi^*, \underline{\pi})} P_\psi^{\pi^*} (r_\psi - \ddot{r}_\psi + \mathbf{V}_{h_\psi}^*(s_\psi) - \ddot{\mathbf{V}}_{h_\psi}^*(s_\psi))$$

$$\geq \sum_{\psi \in \Psi(\pi^*, \underline{\pi})} P_\psi^{\pi^*} (\mathbf{Q}_{h_\psi}^*(s_\psi, \underline{a}) + \text{gap}_{h_\psi}(s_\psi, \underline{a}) - \ddot{\mathbf{V}}_{h_\psi}^*(s_\psi)) \quad (4)$$

$$\geq \sum_{\psi \in \Psi(\pi^*, \underline{\pi})} P_\psi^{\pi^*} (\mathbf{Q}_{h_\psi}^*(s_\psi, \underline{a}) + \frac{1}{2}\text{gap}_{h_\psi}(s_\psi, \underline{a}) - \mathbf{V}_{h_\psi}^{\underline{\pi}}(s_\psi)) \quad (5)$$

$$\geq \frac{1}{2} \sum_{\psi \in \Psi(\pi^*, \underline{\pi})} P_\psi^{\pi^*} (\mathbf{Q}_{h_\psi}^*(s_\psi, \underline{a}) + \text{gap}_{h_\psi}(s_\psi, \underline{a}) - \mathbf{V}_{h_\psi}^{\underline{\pi}}(s_\psi)) \quad (6)$$

$$= \frac{1}{2}(\mathbf{V}_0^* - \mathbf{V}_0^{\underline{\pi}}).$$

(4) results from the fact that $r$ is always larger than or equal to $\ddot{r}$. (5) just makes use of (3). (6) uses $\mathbf{Q}_h^*(s, \underline{a}) \geq \mathbf{Q}_h^{\underline{\pi}}(s, \underline{a}) = \mathbf{V}_h^{\underline{\pi}}(s)$(Lemma A.2). $\qquad\square$

## B  VI-LCB based analysis

### B.1  Algorithm Sketch and Notations

Algorithm used here is Lower Confidence Bound Value Iteration(VI-LCB)[Xie et al., 2021b] with subsampling trick and Berstein-style bonus. The basic idea of LCB is to pessimistically estimate the Q function so that the algorithm won't over estimate some hardly seen suboptimal actions in dataset. The subsampling trick introduced by Li et al. [2022] helps solve the independence problem between $\hat{P}_h$ and $\underline{\mathbf{V}}_{h+1}^{\pi}$, which avoid separating the dataset into H parts, resulting in one H dependency removed in final complexity.

Here we understand dataset as a set of transitions in the form $(h, s, a, s')$ that allows duplicates. When we say that the dataset contains a trajactory $(s_1, a_1, \cdots, s_h, a_h)$, it means that the dataset contains all the decomposed transitions $\{(h, s_h, a_h, s_{h+1})\}_{h=1, \cdots H}$. Also note that $\mathcal{M}$ has deterministic rewards in our setting, so the reward function can be easily derived as long as the $(h, s, a)$ tuple is visited for at least once. And if $(h, s, a)$ is not contained in $\mathcal{D}$, the algorithm output wouldn't be influenced by the value of $r_h(s, a)$, and we can set $r_h(s, a) = 0$. So we assume that the reward function is known from the beginning.

---

**Algorithm 3:** VI-LCB

**input** : Dataset $\mathcal{D}_0$, reward function $r$

1 set $\underline{\mathbf{Q}}_{H+1}^{\pi}(s, a) = 0$
2 set $\underline{\mathbf{V}}_{H+1}^{\pi}(s, a) = 0$
3 **for** $h \leftarrow H$ **to** 1 **do**
4 $\quad$ compute the empirical transition kernel $\hat{P}_h$
5 $\quad$ $\hat{P}_{h,s,a}(s') = \frac{N_h(s,a,s')}{N_h(s,a)}$ with $0/0 = 0$
6 $\quad$ **for** $s \in \mathcal{S}, a \in \mathcal{A}$ **do**
7 $\quad\quad$ $b_h(s, a) \leftarrow C_b \sqrt{\frac{\mathbf{Var}_{\hat{P}_{h,s,a}}(\underline{\mathbf{V}}_{h+1}^{\pi})\iota}{N_h'(s,a)}} + C_b \frac{H\iota}{N_h'(s,a)}$, where $N_h'(s, a) = N_h(s, a) \vee \iota$
8 $\quad\quad$ $\underline{\mathbf{Q}}_h^{\pi}(s, a) \leftarrow \max\{0, r_h(s, a) + \hat{P}_{h,s,a}^{\top} \underline{\mathbf{V}}_{h+1}^{\pi} - b_h(s, a)\}$
9 $\quad$ **for** $s \in \mathcal{S}$ **do**
10 $\quad\quad$ $\underline{\mathbf{V}}_h^{\pi}(s) \leftarrow \max_{a \in \mathcal{A}} \underline{\mathbf{Q}}_h^{\pi}(s, a)$
11 $\quad\quad$ $\underline{\pi}_h(s) \leftarrow \arg\max_{a \in \mathcal{A}} \underline{\mathbf{Q}}_h^{\pi}(s, a)$

**output** : policy $\underline{\pi}$

---

**Algorithm 4:** Subsampled VI-LCB

**input** : Dataset $\mathcal{D}$, reward function $r$

1 Split $\mathcal{D}$ into 2 halves containing same number of sample trajectories, $\mathcal{D}^{\mathrm{main}}$ and $\mathcal{D}^{\mathrm{aux}}$
2 $\mathcal{D}_0 = \{\}$
3 **for** $(h, s) \in [H] \times \mathcal{S}$ **do**
4 $\quad$ $N_h^{\mathrm{trim}}(s) \leftarrow \max\{0, N_h^{\mathrm{aux}}(s) - 10\sqrt{N_h^{\mathrm{aux}}(s) \log \frac{HS}{\delta}}\}$
5 $\quad$ Randomly subsample $\min\{N_h^{\mathrm{trim}}(s), N_h^{\mathrm{main}}(s)\}$ samples of transition from $(h, s)$ from $\quad$ $\mathcal{D}^{\mathrm{main}}$ to add to $\mathcal{D}_0$
6 $\underline{\pi} \leftarrow$ VI-LCB$(\mathcal{D}_0, r)$

**output** : policy $\underline{\pi}$

---

**Notations in VI-LCB.** In the algorithm, $N_h(s, a)$ refers to the number of sample transitions starting from state $s$, taking action $a$ at time step $h$ of some given dataset, and $N_h(s) = \sum_{a \in \mathcal{A}} N_h(s, a)$. Superscripts stand for the dataset. See Li et al. [2022] for a more detailed description of the algorithm.

The proof of independence between samples with different $h$ in $\mathcal{D}_0$ is omitted here, and we will not need it directly because the proof of Lemma B.1 suggests it.

Note that different from the notation in Algorithm 1, we use $\underline{\mathbf{V}}^{\pi}$ and $\underline{\mathbf{Q}}^{\pi}$ instead of $\underline{\mathbf{V}}$ and $\underline{\mathbf{Q}}$ in Algorithm 4. We do this to emphasize that $\underline{\mathbf{V}}$ and $\underline{\mathbf{Q}}$ in Algorithm 1 directly satisfies the definitions of $\underline{\mathbf{V}}^{\pi}$ and $\underline{\mathbf{Q}}^{\pi}$ in thresholding technique (Definition A.2), which will be rigorously proved in Lemma B.3. To avoid unnecessary confusion or reading difficulty, $\underline{\mathbf{V}}$ and $\underline{\mathbf{Q}}$ without superscript stands for the true optimal Q/value functions of $\underline{\mathcal{M}}$, i.e., $\underline{\mathbf{V}}^{\pi}$ and $\underline{\mathbf{Q}}^{\pi}$, in the following proof.

## B.2 Proof Preparation

To warm up, we first prove that VI-LCB perfectly matches our definition of LCB-style algorithm. From the original paper of VI-LCB [Li et al., 2022], we quote a slightly modified version of their lemma 6, where constants and notations are changed, and $V$ is replaced with $\underline{\mathbf{V}}^{\pi}_{h+1}$.

**Lemma B.1** (Transition estimation bound). *For any $1 \leq h \leq H$, with probability at least $1 - \frac{\delta}{2H}$, we have*

$$|(\hat{P}_{h,s,a} - P_{h,s,a})^{\top} \underline{\mathbf{V}}_{h+1}| \leq b_n(s, a) = C_b \sqrt{\frac{\mathbf{Var}_{\hat{P}_{h,s,a}}(\underline{\mathbf{V}}^{\pi}_{h+1})\iota}{N_h(s, a)}} + C_b \frac{H\iota}{N_h(s, a)}, \qquad \forall (s, a) \in \mathcal{S} \times \mathcal{A}.$$

Proof for Lemma B.1 is omitted here. With the union bound, we have the inequality in Lemma B.1 holds for all $h \in [H]$ with probability over $1 - \frac{\delta}{2}$. Also, Lemma 1 from original paper helps with the guarantee of the sample number.

**Lemma B.2.** *With probability over $1 - \frac{\delta}{2}$, we have*

$$N_h(s, a) \geq C_{data}(Nd^{\mu}_h(s, a) - \sqrt{Nd^{\mu}_h(s, a)\iota}), \qquad \forall (h, s, a) \in [H] \times \mathcal{S} \times \mathcal{A},$$

*for some positive constant $C_{data}$.*

Proof of this lemma is also omitted, which is a direct result of Binomial concentration. Then we can prove the concentration lemma, which serves as the basis of following analysis.

**Lemma B.3.** *If we run VI-LCB on a offline learning instance $(\mathcal{M}, \mu)$, with high probability (over $1 - \delta$), the following event $\xi_{conc}$ happens for some positive constant $C_d$:*

1. *the execution instance satisfies the three arguments in Definition A.1, and $0 \leq \mathbf{E}_h(s, a) \leq 2b_h(s, a)$ for all $(h, s, a) \in [H] \times \mathcal{S} \times \mathcal{A}$.*

2. *$N'_h(s, a) = N_h(s, a) \vee \iota \geq C_d Nd^{\mu}_h(s, a)$ for all $(h, s, a) \in [H] \times \mathcal{S} \times \mathcal{A}$.*

*Proof.* We just need to prove that both statements are true with probability over $1 - \frac{\delta}{2}$ respectively, and then we can finish the proof by applying union bound.

**Proof of statement 1:** $\underline{\mathbf{Q}}$ in the definition of pessimistic algorithm matches the $\underline{\mathbf{Q}}$ in VI-LCB. We first prove that $\underline{\mathcal{M}}$ exists. With $\underline{\mathbf{Q}}$ given, we can actually get a closed form of $\underline{r}$,

$$\underline{r}_h(s, a) = \underline{\mathbf{Q}}_h(s, a) - \mathbb{E}_{\underline{\pi}|s_h=h, a_h=a}[\underline{\mathbf{Q}}_h(s_{h+1}, a_{h+1})]$$

$$= \underline{\mathbf{Q}}_h(s, a) - \sum_{s' \in \mathcal{S}} P_{h,s,a}(s') \underline{\mathbf{Q}}_{h+1}(s', \underline{a})$$

$$= \underline{\mathbf{Q}}_h(s, a) - P^{\top}_{h,s,a} \underline{\mathbf{V}}_{h+1}.$$

Then we can find that the definition of $\underline{\mathbf{V}}$ in the algorithm agrees with the one in Definition A.2, and we won't distinguish between these two definitions in following induction. It remains to show that

$$0 \leq \mathbf{E}_h(s, a) \triangleq r_h(s, a) - \underline{r}_h(s, a) \leq 2b_h(s, a). \tag{7}$$

Both inequalities follow from Lemma B.1. Recall that $\underline{\mathbf{Q}}_h(s,a) = \max\{r_h(s,a) + \hat{P}_{h,s,a}^\top \underline{\mathbf{V}}_{h+1} - b_h(s,a), 0\}$,

$$
\begin{aligned}
\underline{\mathbf{Q}}_h(s,a) &= \max\{0, r_h(s,a) + \hat{P}_{h,s,a}^\top \underline{\mathbf{V}}_{h+1} - b_h(s,a)\} \\
&= \max\{0, P_{h,s,a}^\top \underline{\mathbf{V}}_{h+1} + r_h(s,a) + (\hat{P}_{h,s,a} - P_{h,s,a})^\top \underline{\mathbf{V}}_{h+1} - b_h(s,a)\} \\
&\leq \max\{0, P_{h,s,a}^\top \underline{\mathbf{V}}_{h+1} + r_h(s,a)\} = P_{h,s,a}^\top \underline{\mathbf{V}}_{h+1} + r_h(s,a).
\end{aligned}
$$

Simple transformation of above inequality leads to

$$
r_h(s,a) - \underline{r}_h(s,a) \geq \underline{\mathbf{Q}}_h(s,a) - P_{h,s,a}^\top \underline{\mathbf{V}}_{h+1} - \underline{r}_h(s,a) = 0.
$$

The second inequality is also straight forward. We first unfold the definitions of $r$ and $\underline{r}$, then apply Lemma B.1 to get

$$
\begin{aligned}
r_h(s,a) - \underline{r}_h(s,a) &\leq \underline{\mathbf{Q}}_h(s,a) - \hat{P}_{h,s,a}^\top \underline{\mathbf{V}}_{h+1} + b_h(s,a) - (\underline{\mathbf{Q}}_h(s,a) - P_{h,s,a}^\top \underline{\mathbf{V}}_{h+1}) \\
&= b_h(s,a) - (\hat{P}_{h,s,a} - P_{h,s,a})^\top \underline{\mathbf{V}}_{h+1} \\
&\leq 2b_h(s,a).
\end{aligned}
$$

**Proof of statement 2:** We prove over the assumption of event: $N_h(s,a) \geq C_{data}(Nd_h^\mu(s,a) - \sqrt{Nd_h^\mu(s,a)\iota})$ for all $(h,s,a) \in [H] \times \mathcal{S} \times \mathcal{A}$, which is proved by Lemma B.2 to happen with probability over $1 - \frac{\delta}{2}$.

When $Nd_h^\mu(s,a) - \sqrt{Nd_h^\mu(s,a)\iota} \leq \frac{\iota}{C_{data}}$, simple calculation leads to

$$
\sqrt{Nd_h^\mu(s,a)} \leq \frac{1 + \sqrt{1 + \frac{4}{C_{data}}}}{2}\sqrt{\iota} = \lambda\sqrt{\iota},
$$

where $\lambda = \frac{1 + \sqrt{1 + \frac{4}{C_{data}}}}{2}$ is a constant larger than 1. Therefore

$$
N_h(s,a) \vee \iota \geq \iota \geq \frac{1}{\lambda^2} Nd_h^\mu(s,a). \tag{8}
$$

When $Nd_h^\mu(s,a) - \sqrt{Nd_h^\mu(s,a)\iota} \geq \frac{\iota}{C_{data}}$, simple calculation leads to

$$
\begin{aligned}
\sqrt{Nd_h^\mu(s,a)} &\geq \frac{1 + \sqrt{1 + \frac{4}{C_{data}}}}{2}\sqrt{\iota} = \lambda\sqrt{\iota} \\
\Leftrightarrow Nd_h^\mu(s,a) - \sqrt{Nd_h^\mu(s,a)\iota} &\geq (1 - \frac{1}{\lambda})Nd_h^\mu(s,a) \\
\Rightarrow N_h(s,a) \vee \iota \geq C_{data}(Nd_h^\mu(s,a) - \sqrt{Nd_h^\mu(s,a)\iota}) &\geq C_{data}(1 - \frac{1}{\lambda})Nd_h^\mu(s,a). \tag{9}
\end{aligned}
$$

Then together with (8) and (9), and letting $C_d = C_{data}(1 - \frac{1}{\lambda}) \wedge 1$, we finish the proof of statement 2. $\qquad\square$

**Lemma B.4.** *When $0 \leq x \leq y$, for $\epsilon > 0$*

$$
\max\{0, x - \epsilon\} \leq \frac{y^2}{\epsilon}.
$$

*Proof.* When $x \leq \epsilon$, $\max\{0, x - \epsilon\} = 0 \leq \frac{y^2}{\epsilon}$.
When $x > \epsilon$,

$$
\max\{0, x - \epsilon\} \leq y \leq y \cdot \frac{x}{\epsilon} \leq \frac{y^2}{\epsilon}.
$$

$\qquad\square$

## B.3 Proof of Upper Bound with Relative Optimal Policy Coverage(Proof of Theorem 5.1)

This analysis is actually made with Hoeffding bonus for simplicity. Because Berstein bonus is larger than Hoeffding bonus up to log term,

$$b_h(s,a) = C_b\sqrt{\frac{\mathbf{Var}_{\hat{P}_{h,s,a}}(\mathbf{V}^{\pi}_{h+1})\iota}{N'_h(s,a)}} + \frac{C_b H\iota}{N'_h(s,a)} \leq 2C_b\sqrt{\frac{H^2\iota^2}{N'_h(s,a)}}. \tag{10}$$

With Corollary A.1 and Lemma B.4,

$$\mathbf{V}^*_0 - \mathbf{V}^{\pi}_0 \leq 2\sum_{h=1}^{H}\mathbb{E}_{\pi^*}[\max\{0, \mathbf{E}_h(s,a) - \frac{\mathrm{gap}_{\min}}{2H}\}] \qquad\qquad \text{(Corollary A.1)}$$

$$= 2\sum_{h,s} d^*_h(s)\max\{0, \mathbf{E}_h(s,a) - \frac{\mathrm{gap}_{\min}}{2H}\}$$

$$\lesssim \sum_{h,s} d^*_h(s)\frac{b^2_h(s,a^*)}{\frac{\mathrm{gap}_{\min}}{2H}} \qquad\qquad \text{(Lemma B.4 and } \xi_{\mathrm{conc}})$$

$$\lesssim \sum_{h,s} d^*_h(s)\frac{H^3\iota^2}{N_h(s,a^*)'\mathrm{gap}_{\min}}$$

$$\lesssim \sum_{h,s} d^*_h(s)\frac{H^3\iota^2}{Nd^{\mu}_h(s,a)\mathrm{gap}_{\min}} \qquad\qquad (\xi_{\mathrm{conc}})$$

$$\lesssim \frac{1}{N}\sum_{h,s} d^*_h(s)\frac{H^3 C^*\iota^2}{d^*_h(s)\mathrm{gap}_{\min}} \qquad\qquad \text{(relative optimal coverage assumption)}$$

$$= \frac{1}{N}\frac{H^4 S C^*\iota^2}{\mathrm{gap}_{\min}}.$$

Therefore, under relative optimal policy coverage, the sample complexity bound can be

$$N = O(\frac{H^4 S C^*\iota^2}{\epsilon\,\mathrm{gap}_{\min}}).$$

A similar proof in Section B.5 can be applied to prove this result by replacing all the $NP \gtrsim H\iota$ requirements with $N_h(s,a)' \triangleq N_h(s,a) \vee \iota \geq \iota$. So the strict bound without extra $\iota$ can be derived. To avoid redundancy, we omit the proof.

$$N = O(\frac{H^4 S C^*\iota}{\epsilon\,\mathrm{gap}_{\min}}).$$

## B.4 Proofs of Upper Bound with Uniform Optimal Policy Coverage(Proof of Theorem 4.1)

Proof of Theorem 4.1 does not necessarily involve the deficit thresholding technique introduced above. We just need to confirm that $\underline{\mathbf{Q}}^{\pi}_h(s,a^*) \geq \mathbf{V}^*_h(s) - \mathrm{gap}_{\min} \geq \mathbf{V}^*_h(s,a') \geq \underline{\mathbf{Q}}^{\pi}_h(s,a')$, where $a'$ is any suboptimal action, to get a optimal policy. We first present the proof applying this idea, and then present a simpler proof by applying the thresholding technique.

### B.4.1 Proof without Deficit Thresholding Technique

First we introduce a new definition,

$$d^*_{h\sim(h',s')}(s) \triangleq \mathbb{E}_{\pi^*}[\mathbb{I}\{s_h = s\} \mid s_{h'} = s'],$$
$$d^*_{h\sim(h',s')} \triangleq (d^*_{h\sim(h',s')}(s_1), \cdots, d^*_{h\sim(h',s')}(s_S))^{\top} \quad \text{for some certain order of states } s_1, s_2, \cdots s_S.$$

And when there is no confusion, we use $d^{*'}_h$ to denote $d^*_{h\sim(h',s')}$.

**Lemma B.5** (Part Decomposition). $\forall (h', s') \in [H] \times \mathcal{S}$, if the event $\xi_{\text{conc}}$ happens, and $P > 0$, then $\forall$ optimal policy $\pi^*$,

$$\sum_{h=h'}^{H} \sum_{s} d^*_{h \sim (h', s')}(s) b_{h'}(s, a^*) \leq C_e \sqrt{\frac{H^3 \iota}{NP}} + C_e \frac{H^2 \iota}{NP},$$

where $C_e = \max\{4\frac{C_b}{\sqrt{C_d}}, \frac{16C_b^2 + 12C_b}{C_d}, 1\}$.

With this lemma, we can further limit $\underline{\mathbf{Q}}_h^\pi(s, a^*)$.

$$\mathbf{V}_{h'}^*(s') - \underline{\mathbf{Q}}_{h'}^\pi(s', a^*) \leq \mathbf{V}_{h'}^*(s') - \underline{\mathbf{Q}}_{h'}^*(s', a^*) \qquad (\pi \text{ is the optimal policy over } \underline{\mathcal{M}})$$

$$= \sum_{h=h'}^{H} \sum_{s} d^*_{h \sim (h', s')}(s) \mathbf{E}_h(s, a^*)$$

$$\leq 2 \sum_{h,s} d_h^{*'} b_h(s, a^*) \qquad (\xi_{\text{conc}})$$

$$\leq 2C_e \sqrt{\frac{H^3 \iota}{NP}} + 2C_e \frac{H^2 \iota}{NP}. \qquad (\text{Lemma B.5})$$

When $N \geq \frac{4C_e^2 H^3 \iota}{\lambda^2 P}$ for some $\lambda \leq H$,

$$\mathbf{V}_{h'}^*(s') - \underline{\mathbf{Q}}_{h'}^\pi(s', a^*)$$

$$\leq \frac{\lambda}{2} + \frac{\lambda}{2} \frac{\lambda}{2C_e H}$$

$$\leq \lambda.$$

Setting $\lambda = \text{gap}_{\min}$, we get the conclusion that

$$N = \frac{4C_e^2 H^3 \iota}{\text{gap}_{\min}^2 P}$$

can make sure that the returned policy is one of the optimal policies with probability over $1 - \delta$.

### B.4.2 Proof with Deficit Thresholding Technique

By applying the Lemma B.8 which is orginally developed for the proof of Theorem 5.2, we shall directly prove that the suboptimality would be zero if $N > \frac{CH^3 \iota}{P\text{gap}_{\min}}$. Lemma B.8 allows us to set $\epsilon_h(s, a) = C_{pac} \left( \frac{\mathbf{Var}_{\hat{P}_{h,s,a}}(\mathbf{V}_{h+1}^\pi)}{H^2} + \frac{1}{H} \right) \text{gap}_{\min}$. When $N \geq C \frac{H^3 \iota}{P\text{gap}_{\min}^2}$, for any optimal policy $\pi^*$,

$$b_h(s, a^*) = C_b \sqrt{\frac{\mathbf{Var}_{\hat{P}_{h,s,a^*}}(\mathbf{V}_{h+1}^\pi) \iota}{N_h'(s, a^*)}} + C_b \frac{H\iota}{N_h(s, a^*)}$$

$$\lesssim \sqrt{\frac{\mathbf{Var}_{\hat{P}_{h,s,a^*}}(\mathbf{V}_{h+1}^\pi) \iota}{NP}} + \frac{H\iota}{NP} \qquad (\xi_{\text{conc}})$$

$$\leq \sqrt{\frac{\mathbf{Var}_{\hat{P}_{h,s,a^*}}(\mathbf{V}_{h+1}^\pi) \text{gap}_{\min}^2}{CH^3}} + \frac{\text{gap}_{\min}^2}{CH^2} \qquad (N \geq C \frac{H^3 \iota}{P\text{gap}_{\min}^2})$$

$$\leq \frac{\mathbf{Var}_{\hat{P}_{h,s,a^*}}(\mathbf{V}_{h+1}^\pi) \text{gap}_{\min}}{2CH^2} + \frac{\text{gap}_{\min}}{2H} + \frac{\text{gap}_{\min}}{CH} \qquad a + b \geq 2\sqrt{ab}, \ \text{gap}_{\min} \leq H$$

$$\lesssim \epsilon_h(s, a^*).$$

Therefore, with a large enough global constant $C$, we have $\mathbf{E}_h(s, a^*) \leq 2b_h(s, a^*) \leq \epsilon_h(s, a^*)$ holds for any time-state pair. Together with Theorem A.1,

$$\mathbf{V}^* - \mathbf{V}^\pi \leq 2\mathbb{E}_{\pi^*} \left[ \ddot{\mathbf{E}}_h(s, a) \right] = 0.$$

### B.4.3 Tools for the Proof of Lemma B.5

We first introduce a modified version of Lemma from Li et al. [2022]. Note that the proof of this lemma didn't involve any assumption about the data coverage, and is a pure mathmatical analysis. So the original proof is valid, and to avoid redundancy, we won't prove this lemma again.

**Lemma B.6.** $\forall h \in [H]$, and any vector $V \in \mathbb{R}^S$ independent of $\hat{P}_h$ obeying $\|V\|_\infty \leq H$. With probability at least $1 - \delta$, one has

$$
\mathbf{Var}_{\hat{P}_{h,s,a}}(V) \leq 2\mathbf{Var}_{P_{h,s,a}}(V) + \frac{5H^2\iota}{3N_h'(s,a)}
$$

simultaneously for all $(s,a) \in \mathcal{S} \times \mathcal{A}$ obeying $N_h(s,a) > 0$

Modification lies in that we use $N_h'(s,a)$ to replace $N_h(s,a)$ in original version, for when $N_h(s,a) \leq \iota$, the inequalities hold trivially. Also we introduce the lemma needed to limit the overall variance. This lemma differs from Li et al. [2022]'s work from the definition of $d_h^{*'}$ to support our theorem.

**Lemma B.7** (weighted variance sum). $\forall (h', s') \in [H] \times \mathcal{S}$, if the event $\xi_{\mathrm{conc}}$ happens, we have

$$
\sum_{h=h'}^{H} \sum_s d_{h \sim (h',s')}^*(s)\mathbf{Var}_{P_{h,s,a}}(\underline{\mathbf{V}}_{h+1}^\pi) \leq 4H \sum_{h=h'}^{H} \sum_s d_{h \sim (h',s')}^*(s)b_h(s,a^*) + 2H^2.
$$

*Proof.* Here we use $P_h^* \in \mathbb{R}^{S \times S}$ to denote the transition kernel of optimal policy, where $P_{h,(m,n)}^*$ is the probability of transfer from $s_m$ to $s_n$ at step $h$ while applying the optimal policy $\pi^*$. $A \circ B$ refers to the Hadamard product of $A$ and $B$.

$$
\sum_{h=h'}^{H} \sum_s d_h^{*'}(s)\mathbf{Var}_{P_{h,s,a^*}}(\underline{\mathbf{V}}_{h+1}^\pi) = \sum_{h=h'}^{H} d_h^{*'\top}(P_h^* \underline{\mathbf{V}}_{h+1}^\pi \circ \underline{\mathbf{V}}_{h+1}^\pi - (P_h^* \underline{\mathbf{V}}_{h+1}^\pi) \circ (P_h^* \underline{\mathbf{V}}_{h+1}^\pi))
$$

$$
= \sum_{h=h'}^{H} d_h^{*'\top}(P_h^* \underline{\mathbf{V}}_{h+1}^\pi \circ \underline{\mathbf{V}}_{h+1}^\pi - \underline{\mathbf{V}}_h^\pi \circ \underline{\mathbf{V}}_h^\pi + \underline{\mathbf{V}}_h^\pi \circ \underline{\mathbf{V}}_h^\pi - (P_h^* \underline{\mathbf{V}}_{h+1}^\pi) \circ (P_h^* \underline{\mathbf{V}}_{h+1}^\pi))
$$

$$
= \sum_{h=h'}^{H} \left( d_{h+1}^{*'\top} \underline{\mathbf{V}}_{h+1}^\pi \circ \underline{\mathbf{V}}_{h+1}^\pi - d_h^{*'\top} \underline{\mathbf{V}}_h^\pi \circ \underline{\mathbf{V}}_h^\pi \right) + \sum_{h=h'}^{H} d_h^{*'\top}(\underline{\mathbf{V}}_h^\pi \circ \underline{\mathbf{V}}_h^\pi - (P_h^* \underline{\mathbf{V}}_{h+1}^\pi) \circ (P_h^* \underline{\mathbf{V}}_{h+1}^\pi))
$$

$$
= 0 - d_{h'}^{*'\top} \underline{\mathbf{V}}_{h'}^\pi \circ \underline{\mathbf{V}}_{h'}^\pi + \sum_{h=h'}^{H} d_h^{*'\top}(\underline{\mathbf{V}}_h^\pi - P_h^* \underline{\mathbf{V}}_{h+1}^\pi) \circ (\underline{\mathbf{V}}_h^\pi + P_h^* \underline{\mathbf{V}}_{h+1}^\pi)
$$

$$
\leq \sum_{h=h'}^{H} d_h^{*'\top}(\underline{\mathbf{V}}_h^\pi - P_h^* \underline{\mathbf{V}}_{h+1}^\pi) \circ (\underline{\mathbf{V}}_h^\pi + P_h^* \underline{\mathbf{V}}_{h+1}^\pi). \tag{11}
$$

The above induction mainly uses the equality that $d_h^{*'} P_h^* = d_{h+1}^{*'}$ and non-negativity of $d_h^{*'}$. Because the concentration events $\xi_{\mathrm{conc}}$ guarantees that $b_h(s,a) \geq |(\hat{p}_h(s,a) - p_h(s,a))^\top \mathbf{V}_{h+1}|$,

$$
\begin{aligned}
&\underline{\mathbf{V}}_h^\pi(s) - p_h(s,a^*)^\top \underline{\mathbf{V}}_{h+1}^\pi \\
&= \underline{\mathbf{V}}_h^\pi(s) - \underline{\mathbf{Q}}_h^\pi(s,a^*) + r_h(s,a^*) - b_h(s,a^*) + (\hat{p}_h(s,a^*) - p_h(s,a^*))^\top \underline{\mathbf{V}}_{h+1}^\pi \\
&\geq 0 + 0 - b_h(s,a^*) - b_h(s,a^*) = -2b_h(s,a^*).
\end{aligned}
$$

Then we can continue from (11) to get

$$\sum_{h=h'}^{H}\sum_{s}d_h^{*'}(s)\mathbf{Var}_{P_{h,s,a}}(\underline{\mathbf{V}_{h+1}^{\pi}}) \le \sum_{h,s}d_h^{*'\top}(\underline{\mathbf{V}_h^{\pi}} - P_h^*\underline{\mathbf{V}_{h+1}^{\pi}}) \circ (\underline{\mathbf{V}_h^{\pi}} + P_h^*\underline{\mathbf{V}_{h+1}^{\pi}})$$

$$\le \sum_{h=h'}^{H}d_h^{*'\top}(\underline{\mathbf{V}_h^{\pi}} - P_h^*\underline{\mathbf{V}_{h+1}^{\pi}} + 2b_h(s,a^*)\mathbf{1}) \circ (\underline{\mathbf{V}_h^{\pi}} + P_h^*\underline{\mathbf{V}_{h+1}^{\pi}})$$

$$\le 2H\sum_{h=h'}^{H}d_h^{*'\top}(\underline{\mathbf{V}_h^{\pi}} - P_h^*\underline{\mathbf{V}_{h+1}^{\pi}} + 2b_h(s,a^*)\mathbf{1})$$

$$= 2H(d_{h'}^{*'\top}\underline{\mathbf{V}_{h'}^{\pi}} - d_{H+1}^{*'\top}\underline{\mathbf{V}_{H+1}^{\pi}}) + 4H\sum_{h'=h}^{H}\sum_{s}d_{h\sim(h',s')}^{*}(s)b_h(s,a^*)$$

$$\le 2H^2 + 4H\sum_{h'=h}^{H}\sum_{s}d_{h\sim(h',s')}^{*}(s)b_h(s,a^*).$$

$\square$

### B.4.4  Proof of Lemma B.5

*Proof.* This proof is similar to the one in Li et al. [2022]. The difference lies in that we generalize the conclusion to any part decomposition, while the original version only cares about the optimal policy distribution.
First, it follows from Lemma B.6 and inequality $\sqrt{a+b} \le \sqrt{a} + \sqrt{b}$ that

$$\begin{aligned}
\frac{1}{C_b}b_h(s,a) &= \sqrt{\frac{\mathbf{Var}_{\hat{P}_{h,s,a}}(\underline{\mathbf{V}_{h+1}^{\pi}})\iota}{N_h'(s,a)}} + \frac{H\iota}{N_h'(s,a)} \\
&\le \sqrt{\frac{2\mathbf{Var}_{P_{h,s,a}}(\underline{\mathbf{V}_{h+1}^{\pi}})\iota + \frac{5H^2\iota}{3N_h'(s,a)}\iota}{N_h'(s,a)}} + \frac{H\iota}{N_h'(s,a)} \\
&\le \sqrt{\frac{2\mathbf{Var}_{P_{h,s,a}}(\underline{\mathbf{V}_{h+1}^{\pi}})\iota}{N_h'(s,a)}} + (1+\sqrt{\frac{5}{3}})\frac{H\iota}{N_h'(s,a)} \\
&\le 2\sqrt{\frac{\mathbf{Var}_{P_{h,s,a}}(\underline{\mathbf{V}_{h+1}^{\pi}})\iota}{N_h'(s,a)}} + \frac{3H\iota}{N_h'(s,a)}.
\end{aligned}\tag{12}$$

Note that the concentration event $\xi_{\text{conc}}$ guarantees that $N_h(s, a^*) \geq C_d N d_h^\mu(s, a^*) \geq C_d N P$. Then we can use Cauchy-Schwarz Inequality to limit the variance term,

$$\sum_{h,s} d_h^{*'}(s) \sqrt{\frac{\mathbf{Var}_{P_{h,s,a^*}}(\underline{\mathbf{V}}_{h+1}^\pi)\iota}{N_h'(s, a^*)}}$$

$$\leq \sqrt{\frac{\iota}{C_d N P}} \sum_{h,s} d_h^{*'}(s) \sqrt{\mathbf{Var}_{P_{h,s,a^*}}(\underline{\mathbf{V}}_{h+1}^\pi)} \qquad (\xi_{\text{conc}})$$

$$\leq \sqrt{\frac{\iota}{C_d N P}} \sqrt{\sum_{h,s} d_h^{*'}(s)} \sqrt{\sum_{h,s} d_h^{*'}(s) \mathbf{Var}_{P_{h,s,a^*}}(\underline{\mathbf{V}}_{h+1}^\pi)} \qquad \text{(Cauchy-Schwarz's Inequality)}$$

$$\leq \sqrt{\frac{H\iota}{C_d N P}} \sqrt{4H \sum_{h,s} d_h^{*'}(s) b_h(s, a^*) + 2H^2} \qquad \text{(Lemma B.7)}$$

$$\leq \sqrt{\frac{4H^2\iota}{C_d N P} \sum_{h,s} d_h^{*'}(s) b_h(s, a^*)} + \sqrt{\frac{2H^3\iota}{C_d N P}} \qquad (\sqrt{a+b} \leq \sqrt{a} + \sqrt{b})$$

$$\leq \frac{4C_b H^2\iota}{C_d N P} + \frac{1}{2C_b} \sum_{h,s} d_h^{*'}(s) b_h(s, a^*) + \sqrt{\frac{2H^3\iota}{C_d N P}}. \qquad (\sqrt{2ab} \leq a+b)$$

At the same time, we can limit the sum of $\frac{H\iota}{N_h'(s, a^*)}$,

$$\sum_{h,s} d_h^{*'}(s) \frac{H\iota}{N_h'(s, a^*)} \leq \sum_{h,s} d_h^{*'}(s) \frac{H\iota}{C_d N P} = \frac{H^2\iota}{C_d N P}.$$

By connecting these inequalities to (12), we get

$$\sum_{h,s} d_h^{*'}(s) b_h(s, a^*) \leq C_b \sum_{h,s} 2 d_h^{*'}(s) \sqrt{\frac{\mathbf{Var}_{P_{h,s,a}}(\underline{\mathbf{V}}_{h+1}^\pi)\iota}{N_h'(s, a^*)}} + C_b \sum_{h,s} d_h^{*'}(s) \frac{3H\iota}{N_h'(s, a^*)}$$

$$\leq \frac{1}{2} \sum_{h,s} d_h^{*'}(s) b_h(s, a^*) + (8C_b^2 + 6C_b) \frac{H^2\iota}{C_d N P} + 2C_b \sqrt{\frac{H^3\iota}{C_d N P}}.$$

Rearranging the terms, we finish the proof of Lemma B.5. $\qquad\qquad\square$

## B.5  Proof of Upper bound with both assumptions (Proof of Theorem 5.2)

When we have access to both $P$ and $C^*$, we can derive the bound

$$N = O\left(\frac{H^3 S C^* \iota}{\epsilon \text{gap}_{\min}} + \frac{H\iota}{P}\right).$$

To prove this, we need a specially designed $\epsilon_h(s)$ in Theorem A.1. By setting $\epsilon_h(s) = C_{pac}\left(\frac{\mathbf{Var}_{\hat{P}_{h,s,a^*}}(\underline{\mathbf{V}}_{h+1}^\pi)}{H^2} + \frac{1}{H}\right)\text{gap}_{\min}$, we will first prove that $\xi_{gap}$ happenes, and then calculate the suboptimality gap.

### B.5.1  Tools for the Proof of Theorem 5.2

**Lemma B.8.** *If we set* $\epsilon_h(s, a) = C_{pac}\left(\frac{\mathbf{Var}_{\hat{P}_{h,s,a}}(\underline{\mathbf{V}}_{h+1}^\pi)}{H^2} + \frac{1}{H}\right)\text{gap}_{\min}$ *for some small enough constant* $C_{pac}$, *and* $N \geq C_3 \frac{H\iota}{P}$ *for some constant* $C_3$, $\xi_{\text{conc}}$ *indicates* $\xi_{gap}$,

$$\forall (h, s) \in [H] \times \mathcal{S} \qquad \ddot{\mathbf{V}}_h^*(s) \leq \underline{\mathbf{V}}_h^*(s) + \frac{\text{gap}_{\min}}{2}.$$

*Proof.* We have proved in the proof of Corollary A.1 that
$$\sum_{h,s} d_h^{*'}(s)\frac{\text{gap}_{\min}}{4H} \leq \frac{\text{gap}_{\min}}{4},$$
where $d_h^{*'} = d_{h\sim(h',s')}^*$, which is the state distribution of time step $h$ under $\pi^*$ conditioned on having reached $(h', s')$ before. So it remains to show that

$$\sum_{h,s} d_h^{*'}(s)\frac{\text{Var}_{\hat{P}}(\mathbf{V}^{\pi_{h+1}})}{H^2}\text{gap}_{\min} \lesssim \text{gap}_{\min}. \tag{13}$$

This follows from a similar analysis with the proof of uniform optimal policy coverage assumption case.

$$\sum_{h,s} d_h^{*'}(s)\text{Var}_{\hat{P}}(\underline{\mathbf{V}}_{h+1}^{\pi})$$

$$\leq \sum_{h,s} d_h^{*'}(s)\left(\text{Var}_P(\underline{\mathbf{V}}_{h+1}^{\pi}) + \frac{5H^2\iota}{3N_h'(s,a^*)}\right) \qquad (\text{Lemma B.6})$$

$$\lesssim \sum_{h,s} d_h^{*'}(s)\text{Var}_P(\underline{\mathbf{V}}_{h+1}^{\pi}) + H^2 \qquad (N_h'(s,a^*) \geq C_d NP \gtrsim H\iota)$$

$$\leq 4H\sum_{h,s} d_h^{*'}(s)b_h(s,a^*) + 3H^2 \qquad (\text{Lemma B.7})$$

$$\lesssim 4H\left(\sqrt{\frac{H^3\iota}{NP}} + \frac{H^2\iota}{NP}\right) + 3H^2 \qquad (\text{Lemma B.5})$$

$$\lesssim H^2. \qquad (NP \gtrsim H\iota)$$

Then we can finish the proof

$$\ddot{\mathbf{V}}_h^*(s) - \underline{\mathbf{V}}_h^*(s) = \sum_{h,s} d_h^{*'}(s)(\ddot{\mathbf{E}}_h(s,a^*) - \mathbf{E}_h(s,a^*))$$

$$\leq \sum_{h,s} d_h^{*'}(s)\epsilon_h(s)$$

$$\lesssim C_{cap}\sum_{h,s} d_h^{*'}\left(\frac{\text{Var}_{\hat{P}}(\underline{\mathbf{V}}_{h+1}^{\pi})}{H^2} + \frac{1}{H}\right)\text{gap}_{\min}$$

$$\lesssim C_{cap}\text{gap}_{\min}.$$

We can let $C_{cap}$ be small enough to limit the difference between $\ddot{\mathbf{V}}_h^*(s)$ and $\underline{\mathbf{V}}_h^*(s)$ within $\frac{\text{gap}_{\min}}{2}$. $\qquad\square$

**Lemma B.9.**

$$\ddot{\mathbf{E}}_h(s,a^*) \leq 4C_b\sqrt{\frac{\text{Var}_{\hat{P}}(\mathbf{V}_{h+1}^{\pi})\iota}{N_h'(s,a^*)}}\left(\frac{b_h(s,a^*)}{\epsilon_h(s)}\right) + 2C_b\frac{H\iota}{N_h(s,a^*)}.$$

*Proof.* When $\mathbf{E}_h(s,a^*) < \epsilon_h(s)$, $\ddot{\mathbf{E}}_h(s,a^*) = 0$.
When $\mathbf{E}_h(s,a^*) \geq \epsilon_h(s)$,
$$2b_h(s,a^*) \geq \mathbf{E}_h(s,a^*) \geq \epsilon_h(s).$$
$$\Rightarrow \ddot{\mathbf{E}}_h(s,a^*) \leq \mathbf{E}_h(s,a^*)$$
$$\leq 2b_h(s,a^*)$$
$$= 2C_b\sqrt{\frac{\text{Var}_{\hat{P}}(\mathbf{V}_{h+1}^{\pi})\iota}{N_h'(s,a^*)}} + 2C_b\frac{H\iota}{N_h'(s,a^*)}$$
$$\leq 4C_b\sqrt{\frac{\text{Var}_{\hat{P}}(\mathbf{V}_{h+1}^{\pi})\iota}{N_h'(s,a^*)}}\left(\frac{b_h(s,a^*)}{\epsilon_h(s)}\right) + 2C_b\frac{H\iota}{N_h'(s,a^*)}.$$

$\qquad\square$

### B.5.2 Main Proof

*Proof.* We treat the first term in the RHS of Lemma B.9. With basic inequality $a + b \geq 2\sqrt{ab}$, we can first lowe bound $\epsilon_h(s, a)$,

$$\epsilon_h(s, a) = C_{pac} \left( \frac{\mathbf{Var}_{\hat{P}_{h,s,a}}(\mathbf{V}_{h+1}^\pi)}{H^2} + \frac{1}{H} \right) \mathrm{gap}_{\min} \gtrsim \max \left\{ \frac{\mathbf{Var}_{\hat{P}_{h,s,a}}(\mathbf{V}_{h+1}^\pi)}{H^2}, \sqrt{\frac{\mathbf{Var}_{\hat{P}_{h,s,a}}(\mathbf{V}_{h+1}^\pi)}{H^3}} \right\} \mathrm{gap}_{\min}.$$
$$(14)$$

Therefore we have,

$$\sqrt{\frac{\mathbf{Var}_{\hat{P}}(\mathbf{V}_{h+1}^\pi)\iota}{N_h'(s, a^*)}} \left( \frac{b_h(s, a^*)}{\epsilon_h(s, a^*)} \right)$$

$$\lesssim \sqrt{\frac{\mathbf{Var}_{\hat{P}}(\mathbf{V}_{h+1}^\pi)\iota}{N_h'(s, a^*)}} \left( \frac{\sqrt{\frac{\mathbf{Var}_{\hat{P}}(\mathbf{V}_{h+1}^\pi)\iota}{N_h'(s,a^*)}}}{\frac{\mathbf{Var}_{\hat{P}}(\mathbf{V}_{h+1}^\pi)}{H^2}} + \frac{\frac{H\iota}{N_h(s,a)}}{\sqrt{\frac{\mathbf{Var}_{\hat{P}}(\mathbf{V}_{h+1}^\pi)}{H^3}}} \right) \frac{1}{\mathrm{gap}_{\min}}$$

$$= \frac{H^2\iota}{N_h'(s, a^*)\mathrm{gap}_{\min}} + \frac{H^{5/2}\iota^{3/2}}{N_h'^{3/2}(s, a^*)\mathrm{gap}_{\min}}$$

$$\lesssim \frac{2H^2\iota}{N_h'(s, a^*)\mathrm{gap}_{\min}}.$$

The first inequality is gained by expanding $b_h(s, a^*)$ and $\epsilon_h(s, a)$. The second inequality results from the inequality that $N_h'(s, a^*) \geq C_d NP \gtrsim H\iota$. Therefore, we can further write Lemma B.9 as

$$\ddot{\mathbf{E}}_h(s, a^*) \lesssim \frac{H^2\iota}{N_h'(s, a^*)\mathrm{gap}_{\min}}.$$

Then with Lemma B.8, we have event $\xi_{gap}$ hold. Then Theorem A.1 further indicates that for some deterministic optimal policy $\pi^*$,

$$\mathbf{V}_0^* - \mathbf{V}_0^\pi \lesssim \mathbf{V}_0^* - \ddot{\mathbf{V}}_0^*$$
$$= \sum_{h,s} d_h^*(s)\ddot{\mathbf{E}}_h(s, a^*)$$
$$\lesssim \sum_{h,s} \frac{d_h^*(s)H^2\iota}{N_h'(s, a^*)\mathrm{gap}_{\min}}$$
$$\lesssim \sum_{h,s} \frac{H^2 C^*\iota}{N\mathrm{gap}_{\min}}$$
$$= \frac{H^3 C^* S\iota}{N\mathrm{gap}_{\min}} \lesssim \epsilon.$$

$\square$

## C   Gap-dependent Lower Bounds

We begin by restating the formal version of lower bounds.

**Definition C.1** (offline learning algorithm). *For an algorithm **ALG**, we call it an offline learning algorithm if*

    *1.* **ALG** *takes a dataset $\mathcal{D}$ and optionally a reward function $R$ as input,*

    *2.* **ALG** *output a valid policy $\pi$.*

Notice that **ALG** can be stochastic.

## C.1 Main Results

**Theorem C.1.** *There exists constant $C_{lb}$, s.t. for any $A \geq 3, S \geq 2, H \geq 2, \tau < \frac{1}{2}, \lambda < \frac{1}{3}, \lambda_1 \geq 2$ and algorithm* **ALG***, if the number of sample trajectories*

$$N \leq C_{lb} \cdot \frac{HS\lambda_1}{\lambda \tau^2},$$

*there exists some MDP $\mathcal{M}$ and behavior policy $\mu$ with $\mathrm{gap}_{\min} = \tau, P \geq \frac{\lambda}{eS\lambda_1}, C^* \leq \lambda_1$ such that the output policy $\hat{\pi}$ suffers from a expected suboptimality*

$$\mathbb{E}_{\mathcal{M},\mu,\mathbf{ALG}}[\mathbf{V}_0^* - \mathbf{V}_0^{\hat{\pi}}] \geq \frac{\lambda H \tau}{12}.$$

**Corollary C.1** (lower bound for uniform optimal policy coverage)**.** *Given $A \geq 3, S \geq 2, H \geq 3, P \in (0, \frac{1}{6S}], \epsilon < 1/12, \mathrm{gap}_{\min} \in [\frac{24\epsilon}{H}, \frac{1}{2}]$ and any offline learning algorithm* **ALG** *returning a policy $\hat{\pi}$, there exists a constant $C_1$ such that if the number of offline sample trajectories*

$$N \leq C_1 \cdot \frac{H}{P\mathrm{gap}_{\min}^2},$$

*then there exists a MDP instance $M$ and behavior policy $\mu$ such that the output policy $\hat{\pi}$ suffers from expected $\epsilon$-suboptimality*

$$\mathbb{E}_{\mathcal{M},\mu,\mathbf{ALG}}[\mathbf{V}_0^* - \mathbf{V}_0^{\hat{\pi}}] \geq \epsilon.$$

*Proof.* Let $\lambda = 1/3, \lambda_1 = \frac{1}{3PS}$ and $\tau = \mathrm{gap}_{\min}$ in Theorem C.1, we get the proposition. □

**Corollary C.2** (lower bound for relative optimal policy coverage)**.** *Given $A \geq 3, S \geq 2, H \geq 2, C^* \geq 2, \epsilon < 1/12, \mathrm{gap}_{\min} \in [\frac{24\epsilon}{H}, \frac{1}{2}]$ and any offline learning algorithm* **ALG** *returning a policy $\hat{\pi}$, there exists a constant $C_2$ such that if the number of offline sample trajectories*

$$N \leq C_2 \cdot \frac{H^2 SC^*}{\mathrm{gap}_{\min}\epsilon},$$

*then there exists a MDP instance $\mathcal{M}$ and behavior policy $\mu$ such that the output policy $\hat{\pi}$ suffers from expected $\epsilon$-suboptimality*

$$\mathbb{E}_{\mathcal{M},\mu,\mathbf{ALG}}[\mathbf{V}_0^* - \mathbf{V}_0^{\hat{\pi}}] \geq \epsilon.$$

*Proof.* Let $\lambda = \frac{12\epsilon}{H\mathrm{gap}_{\min}}, \lambda_1 = C^*$ and $\tau = \mathrm{gap}_{\min}$ in Theorem C.1, we get the conclusion. □

## C.2 Proof of Theorem C.1

### C.2.1 Construction of the MDP Family and Behavior Policy

We construct a MDP family and calculate the average minimum suboptimality.
First, we construct the prototype MDP $\mathcal{M}_0$ with $S + 2$ states, horizon of $2H + 1$ and $A$ actions. There are 3 kind of states

1. good state $s_g$. An absorbing state. Reaching this state means a total reward of $H$.

2. bad state $s_b$. An absorbing state. Reaching this state means a total reward of $0$.

3. true states $s_1, s_2, \cdots, s_S$. Actions chosen in these states determine the probability being transfered to $s_g$ and $s_b$.

The initial state distribution $p_0(s)$ is

$$p_0(s) = \begin{cases} \frac{\lambda}{S} & s \in \{s_1, s_2, \cdots, s_S\}, \\ \frac{1-\lambda}{2} & s = s_b, \\ \frac{1-\lambda}{2} & s = s_g. \end{cases}$$

For any $\lambda \in [0, \frac{1}{3}]$. The avaliable action set is $\{a_i\}_{i=1}^A$. The only non-zero rewards in this MDP are $r_h(s_g, a) = 1$ for $h \geq H + 2$ and any $a$. The transition probability of $\mathcal{M}_0$ in the first $H + 1$ steps is,

$$p_h(s_i, a_j, s_i) = 1 - \frac{1}{H} \qquad \forall (h, i, j) \in [H] \times [S] \times [A],$$

$$p_h(s_i, a_j, s_g) = p_h(s_i, a_j, s_b) = \frac{1}{2H} \qquad \forall (h, i, j) \in [H] \times [S] \times [A],$$

$$p_{H+1}(s_i, a_j, s_g) = p_{H+1}(s_i, a_j, s_b) = \frac{1}{2} \qquad \forall (i, j) \in [S] \times [A].$$

For all the other $(h, s, a)$ tuples not mentioned, $p_h(s, a, s) = 1, p_h(s, a, s') = 0$, where $s'$ is any state other than $s$.

Then we construct the MDP family $M$ on the basis of $\mathcal{M}_0$. For each matrix $\phi \in [1, 2]^{H \times S}$, we define $\mathcal{M}_\phi$ to be the MDP almost the same as $\mathcal{M}_0$ except for that

$$p_h(s_i, a_{\phi_{h,i}}, s_g) = \frac{1}{2H}(1 + 2\tau),$$

$$p_h(s_i, a_{\phi_{h,i}}, s_b) = \frac{1}{2H}(1 - 2\tau).$$

In other words, we make the action $a_{\Phi_{h,i}}$ the unique optimal action by lifting it's expected reward by $\tau$. The behavior policy $\mu$ chooses $a_1, a_2$ with probability $1/\lambda_1$ respectively and choose $a_3$ with probability $1 - 2/\lambda_1$ at $\{s_i \mid i = 1, 2, \cdots S\}$, and always choose $a_1$ at $s_g$ and $s_b$. We will prove the following lemmas in Section C.2.4

**Lemma C.1.** *For any MDP constructed above and $\mu$, we have both assumptions hold with*

$$C^* = \lambda_1, P \geq \frac{\lambda}{eS\lambda_1}, \mathrm{gap}_{\min} = \tau.$$

**Lemma C.2.** *For a given algorithm* **ALG***, define the expectation of mistakes made by $\hat{\pi}$ at step $h$, state $s_i$ over the uniform distribution $\nu$ of $\phi$ to be*

$$l_{h,i}(\mathbf{ALG}) = \mathbb{E}_{\mathcal{D} \sim (\mathcal{M}_\phi, \mu), \mathbf{ALG}}[\mathbb{I}\{\hat{\pi}_h(s_i) \neq a_{\phi_{h,i}}\}].$$

*Then expected suboptimality with respect to the randomness of $\mathcal{M}_\phi$ and $\mu$ can be lower bounded by*

$$\mathbb{E}_{\mathcal{M}_\phi, \mu, \mathbf{ALG}}[\mathbf{V}_0^* - \mathbf{V}_0^{\hat{\pi}}] \geq \frac{\lambda}{eS}\tau \sum_{(h,i) \in [H] \times [S]} l_{h,i}(\mathbf{ALG}).$$

**Lemma C.3.** *For any MDP constructed above, we have,*

$$\max_\pi \mathbf{V}_0^* - \mathbf{V}_0^\pi \leq \lambda H \tau.$$

### C.2.2  Main Proof

*Proof.* To avoid making the proof prolix, we strengthen **ALG** by letting **ALG** know that the only thing influencing the value function of a state is the probabilities of transferring to $s_g$ and $s_b$, and the total reward after getting to $s_g$ is exactly $H$, which assumption is conventionally made for the lower bound proofs in MDPs. In this setting, any reasonable algorithm **ALG** would only consider the visitation counts $N_{h,i} = \{N_h(s_i, a, s') \mid a \in \mathcal{A}, s' \in \mathcal{S}\}$ at step $h$ when determining the value of $\hat{\pi}_h(s_i)$.

Then we can rewrite $\bar{l}_{h,i}(\mathbf{ALG}) \triangleq \mathbb{E}_{\phi \sim \nu}[l_{h,i}(\mathbf{ALG})]$,

$$\bar{l}_{h,i}(\mathbf{ALG}) = \mathbb{E}_{\phi \sim \nu} \mathbb{E}_{\mathcal{D} \sim (\mathcal{M}_\phi, \mu), \mathbf{ALG}}[\mathbb{I}\{\hat{\pi}_h(s_i) \neq a_{\phi_{h,i}}\}]$$

$$= \mathbb{E}_{\phi \sim \nu} \mathbb{E}_{N_{h,i} \sim (\mathcal{M}_\phi, \mu), \mathbf{ALG}}[\mathbb{I}\{\hat{\pi}_h(s_i) \neq a_{\phi_{h,i}}\}].$$

Because the KL divergence between the transition kernel $P_{h,s_i,a}^{\mathcal{M}_0}$ and $P_{h,s_i,a_{\phi_{h,i}}}^{\mathcal{M}_\phi}$ satisfies

$$KL\left(\left(1 - \frac{1}{H}, \frac{1}{2H}, \frac{1}{2H}\right) \,\middle\|\, \left(1 - \frac{1}{H}, \frac{1}{H}(\frac{1}{2} + 2\tau), \frac{1}{H}(\frac{1}{2} - 2\tau)\right)\right)$$

$$= \frac{1}{2H} \log \frac{1}{1 - 4\tau^2} \leq \frac{4\tau^2}{H}. \tag{15}$$

We have

$$\bar{l}_{h,i}(\mathbf{ALG}) = \mathbb{E}_{\phi\sim\nu}\mathbb{E}_{N_{h,i}\sim(\mathcal{M}_\phi,\mu),\mathbf{ALG}}[\mathbb{I}\{\hat{\pi}_h(s_i)\neq a_{\phi_{h,i}}\}]$$

$$\geq \mathbb{E}_{\phi\sim\nu}\mathbb{E}_{N_{h,i}\sim(\mathcal{M}_0,\mu),\mathbf{ALG}}[\mathbb{I}\{\hat{\pi}_h(s_i)\neq a_{\phi_{h,i}}\}]$$

$$\quad - \mathbb{E}_{\phi\sim\nu}[\mathrm{TV}(N_{h,i}\mid_{\mathcal{M}_0,\mu}, N_{h,i}\mid_{\mathcal{M}_\phi,\mu})]$$

$$\geq \mathbb{E}_{N_{h,i}\sim(\mathcal{M}_0,\mu),\mathbf{ALG}}\mathbb{E}_{\phi\sim\nu}[\mathbb{I}\{\hat{\pi}_h(s_i)\neq a_{\phi_{h,i}}\}]$$

$$\quad - \mathbb{E}_{\phi\sim\nu}\sqrt{\frac{1}{2}\mathrm{KL}(N_{h,i}\mid_{\mathcal{M}_0,\mu}\|N_{h,i}\mid_{\mathcal{M}_\phi,\mu})} \qquad \text{(Pinsker's inequality)}$$

$$\geq \mathbb{E}_{N_{h,i}\sim(\mathcal{M}_0,\mu)}[\frac{1}{2}] \qquad\qquad (a_1 \text{ and } a_2 \text{ can not be distinguished in } \mathcal{M}_0)$$

$$\quad - \mathbb{E}_{\phi\sim\nu}\sqrt{\frac{1}{2}\sum_{a\in\mathcal{A}}\mathbb{E}_{\mathcal{M}_0,\mu}[N_h(s_i,a)]\mathrm{KL}(P^{\mathcal{M}_0}_{h,s_i,a}\|P^{\mathcal{M}_\phi}_{h,s_i,a})} \quad \text{(KL decomposition)}$$

$$= \frac{1}{2} - \mathbb{E}_{\phi\sim\nu}\sqrt{\frac{1}{2}\mathbb{E}_{\mathcal{M}_0,\mu}[N_h(s_i,a)]\mathrm{KL}(P^{\mathcal{M}_0}_{h,s_i,a_{\phi_{h,i}}}\|P^{\mathcal{M}_\phi}_{h,s_i,a_{\phi_{h,i}}})}$$

$$\geq \frac{1}{2} - \mathbb{E}_{\phi\sim\nu}\sqrt{\mathbb{E}_{\mathcal{M}_0,\mu}[N_h(s_i,a_{\phi_{h,i}})]\frac{2\tau^2}{H}} \qquad \text{(statement (15))}$$

$$\geq \frac{1}{2} - \sqrt{\frac{1}{2}\sum_{a=a_1,a_2}\mathbb{E}_{\mathcal{M}_0,\mu}[N_h(s_i,a)]\frac{2\tau^2}{H}} \qquad \text{(Jensen's inequality)}$$

$$= \frac{1}{2} - \sqrt{\mathbb{E}_{\mathcal{M}_0,\mu}[N_h(s_i)]\frac{2\tau^2}{H\lambda_1}}.$$

$$(\frac{\sum_{a=a_1,a_2}\mathbb{E}_{\mathcal{M}_0,\mu}[N_h(s_i,a)]}{\mathbb{E}_{\mathcal{M}_0,\mu}[N_h(s_i)]} = \mu_h(a_1\mid s_i) + \mu_h(a_2\mid s_i) = \frac{2}{\lambda_1})$$

This further indicates that the expectation of overall mistakes can be lower bounded by

$$\sum_{(h,i)\in[H]\times[S]}\bar{l}_{h,i}(\mathbf{ALG}) \geq \sum_{h,i}\left(\frac{1}{2} - \sqrt{\mathbb{E}_{\mathcal{M}_0,\mu}[N_h(s_i)]\frac{2\tau^2}{H\lambda_1}}\right)$$

$$\geq \frac{HS}{2} - \sqrt{HS}\sqrt{\sum_{h,i}\mathbb{E}_{\mathcal{M}_0,\mu}[N_h(s_i)]\frac{2\tau^2}{H\lambda_1}}$$

$$\text{(Cauchy Schwarz's Inequality)}$$

$$= HS(\frac{1}{2} - \sqrt{\mathbb{E}_{\mathcal{M}_0,\mu}[\sum_{h,i}N_h(s_i)]\frac{2\tau^2}{H^2 S\lambda_1}}). \qquad (16)$$

Because state $s_i$ can be reached at step $h$ only when the initial state is $s_i$,

$$\mathbb{E}_{\mathcal{M}_0,\mu}[\sum_{(h,i)\in[H]\times[S]}N_h(s_i)] = N\sum_{h,i}\frac{\lambda}{S}(1-\frac{1}{H})^{h-1}$$

$$\leq N\sum_{h,i}\frac{\lambda}{S} = NH\lambda.$$

Therefore, continue from inequality (16),

$$\sum_{(h,i)\in[H]\times[S]}\bar{l}_{h,i}(\mathbf{ALG}) \geq HS\left(\frac{1}{2} - \sqrt{N\frac{2\lambda\tau^2}{HS\lambda_1}}\right),$$

Now we can lower bound the suboptimality of $\hat{\pi}$ with Lemma C.2 by

$$\mathbb{E}_{\phi\sim\nu}\mathbb{E}_{\mathcal{M}_\phi,\mu,\mathbf{ALG}}[\mathbf{V}_0^* - \mathbf{V}_0^{\hat{\pi}}] \geq \mathbb{E}_{\phi\sim\nu}[\frac{\lambda\tau}{eS}\sum_{h,i}l_{h,i}(\mathbf{ALG})] \geq \frac{\lambda H\tau}{e}\left(\frac{1}{2} - \sqrt{N\frac{2\lambda\tau^2}{HS\lambda_1}}\right).$$

Then we reach the conclusion that when

$$N \leq \frac{HS\lambda_1}{32\lambda\tau^2},$$

the average suboptimality of $\hat{\pi}$ must be large

$$\mathbb{E}_{\phi\sim\nu}\mathbb{E}_{\mathcal{M}_\phi,\mu,\mathbf{ALG}}[\mathbf{V}_0^* - \mathbf{V}_0^{\hat{\pi}}] \geq \frac{\lambda H\tau}{e} \cdot (\frac{1}{2} - \frac{1}{4}),$$

$$\Rightarrow \exists\phi, s.t. \mathbb{E}_{\mathcal{M}_\phi,\mu,\mathbf{ALG}}[\mathbf{V}_0^* - \mathbf{V}_0^{\hat{\pi}}] \geq \frac{\lambda H\tau}{4e}.$$

$\square$

### C.2.3 Constant probability version of main theorem

Theorem C.1 is stated in the form of expectation, which is not directly consist with upper bound. Here we restate it in the language of probability,

**Theorem C.2.** *There exists constant $C_{lb}$, s.t. for any $A \geq 3, S \geq 2, H \geq 2, \tau < \frac{1}{2}, \lambda < \frac{1}{3}, \lambda_1 \geq 2$ and algorithm* $\mathbf{ALG}$*, if the number of sample trajectories*

$$N \leq C_{lb} \cdot \frac{HS\lambda_1}{\lambda\tau^2},$$

*there exists some MDP $\mathcal{M}$ and behavior policy $\mu$ with* $\mathrm{gap}_{\min} = \tau$, $P \geq \frac{\lambda}{eS\lambda_1}$, $C^* \leq \lambda_1$ *such that the output policy $\hat{\pi}$ suffers from a expected suboptimality*

$$\mathbf{V}_0^* - \mathbf{V}_0^{\hat{\pi}} \geq \frac{\lambda H\tau}{24},$$

*with a probability over $\frac{1}{24}$.*

*Proof.* From the last line of the proof of Theorem C.1, we know that there exists a MDP $\mathcal{M}$, such that

$$\mathbb{E}_{\mathcal{M}_\phi,\mu,\mathbf{ALG}}[\mathbf{V}_0^* - \mathbf{V}_0^{\hat{\pi}}] \geq \frac{\lambda H\tau}{12},$$

and it follows from Lemma C.3 that the random variable $\mathbf{V}_0^* - \mathbf{V}_0^{\hat{\pi}} \leq \lambda H\tau$. Therefore

$$\frac{\lambda H\tau}{12} \leq \mathbb{E}_{\mathcal{M}_\phi,\mu,\mathbf{ALG}}[\mathbf{V}_0^* - \mathbf{V}_0^{\hat{\pi}}]$$

$$\leq \lambda H\tau\mathbb{P}[\mathbf{V}_0^* - \mathbf{V}_0^{\hat{\pi}} > \frac{\lambda H\tau}{24}] + \frac{\lambda H\tau}{24}\mathbb{P}[\mathbf{V}_0^* - \mathbf{V}_0^{\hat{\pi}} \leq \frac{\lambda H\tau}{24}]$$

$$\leq \frac{\lambda H\tau}{12}(12\mathbb{P}[\mathbf{V}_0^* - \mathbf{V}_0^{\hat{\pi}} > \frac{\lambda H\tau}{24}] + \frac{1}{2})$$

$$\Rightarrow \mathbb{P}[\mathbf{V}_0^* - \mathbf{V}_0^{\hat{\pi}} > \frac{\lambda H\tau}{24}] \geq \frac{1}{24}$$

$\square$

### C.2.4 Proof of Lemma C.1

*Proof.* From the construction we see that the policy doesn't influence the probability of reaching $s_i$ at any time step. And the uniform random behavior policy makes sure that there is a chance of $1/A$ to visit action $a_i$ at any state. For $s_g$ and $s_b$, because each of them has a initial probability of $1/3$, the probability of reaching one of them at ant time step would be in $[\frac{1}{3}, \frac{2}{3}]$. By letting a optimal policy always choose $a_1$ in $s_g$ and $s_b$ as $\mu$ does, we make $\frac{d_h^*(s_g)}{d_h^\mu(s_g)} \leq \frac{2/3}{1/3} = 2 \leq C^*$. Therefore $C^* = \lambda_1$
As for P, we see that the probability to reach $s_g$ and $s_d$ with behavior policy at step $h$ is

$$d_h^\mu(s_g, a) \geq \frac{1-\lambda}{2} \geq \frac{\lambda}{2} \geq \frac{\lambda}{eS\lambda_1},$$

$$d_h^\mu(s_b, a) \geq \frac{1-\lambda}{2} \geq \frac{\lambda}{2} \geq \frac{\lambda}{eS\lambda_1},$$

$$d_h^\mu(s_i, a) = \frac{\lambda}{S}(1 - \frac{1}{H})^{h-1}\frac{1}{\lambda_1} \geq \frac{\lambda}{eS\lambda_1}.$$

The part for $\text{gap}_{\min}$ is direct calculation,

$$\text{gap}_{\min} = \frac{1}{2H}(1 + 2\tau)H + 0 - \frac{1}{2H}H - 0 = \tau.$$

$\square$

### C.2.5 Proof of Lemma C.2

*Proof.* Because $\hat{\pi}$ only makes mistakes in $s_i$, and each mistake results in a expected $\tau$ decrease in final cumulative reward, we can directly calculate the expected loss with the performance difference lemma for finite-horizon MDP ,

$$\mathbb{E}_{\mathcal{M}_\phi, \mu, \mathbf{ALG}}[\mathbf{V}_0^* - \mathbf{V}_0^{\hat{\pi}}] = \mathbb{E}_{\hat{\pi}, \mathcal{M}_\phi}[\text{gap}_h(s_h, a_h)]$$

$$= \sum_{i=1}^{S} \sum_{h=1}^{H} d_h^\mu(s_i) l_{h,i}(\mathbf{ALG}) \tau$$

$$= \sum_{i=1}^{S} \sum_{h=1}^{H} P_0(s_i)(1 - \frac{1}{H})^{h-1} \tau l_{h,i}(\mathbf{ALG})$$

$$\geq \sum_{i=1}^{S} \sum_{h=1}^{H} \frac{\lambda}{eS} \tau l_{h,i}(\mathbf{ALG})$$

$$\geq \frac{\lambda}{eS} \tau \sum_{(h,i) \in [H] \times [S]} l_{h,i}(\mathbf{ALG}).$$

$\square$

### C.2.6 Proof of Lemma C.3

*Proof.* The proof is a direct result of performance decomposition lemma.

$$\max_\pi \mathbf{V}_0^* - \mathbf{V}_0^\pi = \max_\pi \sum_{h=1}^{2H} \mathbb{E}_{\pi^*}[\mathbf{V}_h^*(s) - \mathbf{V}_h^\pi(s)]$$

$$= \max_\pi \sum_{h=1}^{H} \mathbb{E}_{\pi^*}[\mathbf{V}_h^*(s) - \mathbf{V}_h^\pi(s)]$$

$$\leq \sum_{h=1}^{H} \mathbb{E}_{\pi^*}[\max_\pi \mathbf{V}_h^*(s) - \mathbf{V}_h^\pi(s)]$$

$$= \sum_{h=1}^{H} \sum_{i=1}^{S} d_h^*(s) \max_\pi \mathbf{V}_h^*(s_i) - \mathbf{V}_h^\pi(s_i)$$

$$= \sum_{h=1}^{H} \sum_{i=1}^{S} d_h^*(s) \tau$$

$$= \lambda H \tau$$

$\square$

## D  Proof of Necessity of Overall Data Coverage

One may wonder if the Assumption 3.1 has been too strong, because the minimax bound $O(\frac{H^3 \iota}{P\epsilon^2})$ only requires the data coverage over a single optimal policy. Here we give a proof that to derive $\epsilon$-irrevelant bounds for Algorithm 2, single optimal policy coverage is not sufficient.
We provide a hard instance to prove that if we only have data coverage over one of the optimal

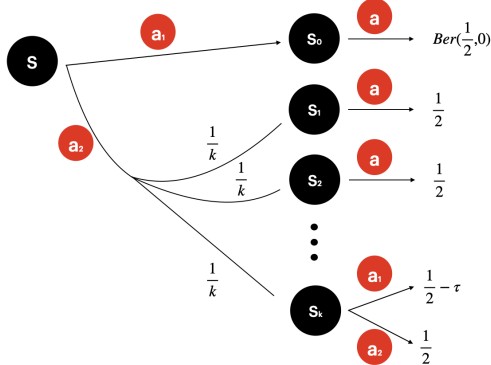

Figure 1: A hard instance with horizon 2, 2 actions and $k + 2$ states. $a_1$ at $s$ leads to $s_0$. The reward of both actions at $s_0$ is sampled from Bernoulli Distribution. $a_2$ at $s$ leads to a uniformly random transition to $s_i, i = 1, \cdots, k$. The reward of both actions at $s_i, i = 1, \cdots, k - 1$ are $\frac{1}{2}$. $a_1$ at $s_k$ receives $\frac{1}{2} - \tau$ reward and $a_2$ at $s_k$ receives $\frac{1}{2}$ reward. $\text{gap}_{\min} = \tau$ in this MDP.

policies, Algorithm 2 may output suboptimal policy with probability over $1/2$. We use $P'$ to refer to the single policy coverage coefficient, i.e.,

$$P' = \max_{\pi^*} \min_{d_h^{\pi^*}(s,a) > 0} d_h^\mu(s, a).$$

Here we consider the MDP with horizon length 2, 2 actions and k+2 states, which is illustrated in Figure 1. The initial state is $s$, and $P(s, a_1, s_0) = 1, P(s, a_2, s_i) = \frac{1}{k}$ for $i = 1, 2 \cdots, k$. The rewards of actions in $s_i, i = 1, 2 \cdots, k - 1$ are all $\frac{1}{2}$, and the rewards of both actions in $s_0$ are sampled from $\mathcal{N}(\frac{1}{2}, 1)$. $r(s_k, a_1) = \frac{1}{2} - \tau, r(s_k, a_2) = \frac{1}{2}$.

In this MDP $\text{gap}_{\min} = \epsilon$, and the only suboptimal action is to take $a_2$ at $s_k$. Define $\mu$,

$$\mu(a_1 \mid s) = \frac{1}{k + 1},$$
$$\mu(a_2 \mid s) = \frac{k}{k + 1},$$
$$\mu(a_1 \mid s_i) = 1 \qquad i = 0, 1, \cdots, k,$$
$$\mu(a_2 \mid s_i) = 0 \qquad i = 0, 1, \cdots, k.$$

Then we can see that an optimal policy/route $s - a_1 - s_0 - a_1$ has been covered by $\mu$ with minimal coverage distribution $P' = \frac{1}{k+1}$. Then we show that for any constant $C$, the output policy of VI-LCB with $N = \frac{C}{P' \text{gap}_{\min}^2} = \frac{C(k+1)}{\tau^2}$ sample trajectories cannot be guaranteed to be optimal with high probability. For the conciseness of proof, we assume that $C > 10, C_b > 16$ and let $k > 10$.

Intuitively, this is because there exists the probability that some not-so-well covered optimal policy outperforms the covered one in execution process, and no optimality can be guaranteed over the not-so-well covered one. In this instance, $(s, a_2)$ is also optimal, but as no information about $s_k, a_2$ is known by VI-LCB, it will choose $a_1$ following the principle of pessimism.

In the following proof, we omit the subscripts indicating the time step because the each state only appears in time step 1 or 2, which will not incur confusion. Rigorously, we define the event the $\{\hat{\pi}(s) = a_2\}$ as $\xi_{\text{bad}}$,

$$\mathbb{P}[\xi_{bad}] = \mathbb{P}[\underline{\mathbf{Q}}(s, a_1) \leq \underline{\mathbf{Q}}(s, a_2)]$$
$$\geq \mathbb{P}[\underline{\mathbf{Q}}(s, a_1) \leq \frac{1}{2} - \lambda\tau \leq \underline{\mathbf{Q}}(s, a_2)]$$
$$\geq 1 - \mathbb{P}[\underline{\mathbf{Q}}(s, a_1) \geq \frac{1}{2} - \lambda\tau] - \mathbb{P}[\underline{\mathbf{Q}}(s, a_2) \leq \frac{1}{2} - \lambda\tau].$$

where $\lambda$ can be any positive constant, which will be determined later. We limit these two terms respectively.

$$\mathbb{P}[\underline{\mathbf{Q}}(s, a_1) \geq \frac{1}{2} - \lambda\tau] \leq \mathbb{P}[N(s, a_1) \geq \frac{C_1 N}{k+1}] + \mathbb{P}[N(s, a_1) \leq \frac{C_2 N}{k+1}]$$
$$+ \mathbb{P}[N(s, a_1) \in [\frac{C_2 C}{\tau^2}, \frac{C_1 C}{\tau^2}], \ \underline{\mathbf{Q}}(s, a_1) \geq \frac{1}{2} - \lambda\tau].$$

Because $N(s, a_1) \sim Bio(N, \frac{1}{k+1})$, it follows from the asymptotic feature of binomial distribution that $N(s, a_1) - \frac{C}{\tau^2} \sim subG(\frac{Ck}{\tau^2(k+1)})$, and then

$$\mathbb{P}[N(s, a_1) \geq \frac{C_1 C}{\tau^2}] \leq \exp\left(-\frac{(C_1 - 1)^2 C^2}{2(1 - 1/(k+1))^2}\right) \leq \exp(-\frac{C^2}{4}(C_1 - 1)^2),$$
$$\mathbb{P}[N(s, a_1) \leq \frac{C_2 C}{\tau^2}] \leq \exp\left(-\frac{(C_2 - 1)^2 C^2}{2(1 - 1/(k+1))^2}\right) \leq \exp(-\frac{C^2}{4}(C_2 - 1)^2).$$

Let $C_1 = 1.5, C_2 = 0.5$. Remember that we assume that $C > 10$, and this makes the sum of above two terms a small constant smaller than 0.1. Because $\underline{\mathbf{Q}}(s, a_1) = -b(s, a_1) + \hat{r}(s_0, a_1) - b(s, a_1) \leq -\frac{3C_b\sqrt{\iota}}{\sqrt{N(s,a_1)}} + \hat{r}(s_0, a_1)$, and the center limit theorem allow us to use $X \sim \mathcal{N}(\frac{1}{2}, \frac{1}{N(s_0,a_1)})$ to replace $\hat{h}(s_0, a_1)$,

$$\mathbb{P}[N(s, a_1) \in [\frac{C_2 C}{\tau^2}, \frac{C_1 C}{\tau^2}], \ \underline{\mathbf{Q}}(s, a_1) \geq \frac{1}{2} - \lambda\tau] \leq \mathbb{P}[\hat{r}(s_0, a_1) - \frac{3C_b\sqrt{\iota}\tau}{\sqrt{C_1 C}} \geq \frac{1}{2} - \lambda\tau]$$
$$\lesssim \mathbb{P}[X - \frac{1}{2} \geq \frac{3C_b\sqrt{\iota}\tau}{\sqrt{C_1 C}} - \lambda\tau]$$
$$\leq \exp(-\frac{1}{2}(\frac{3C_b\sqrt{\iota}}{\sqrt{C_1 C}} - \lambda)^2 C C_2)$$
$$\leq \exp(-(\frac{1}{2}C_b\sqrt{\iota} - \lambda)^2).$$

The above term disappears quickly when $\iota = \Omega(\log\frac{k}{\delta})$ becomes larger. We will choose a $\lambda < \frac{1}{4}C_b\sqrt{\iota}$. Then we consider $\underline{\mathbf{Q}}(s, a_2)$,

$$\underline{\mathbf{Q}}(s, a_2) = -b(s, a_2) - \sum_{i=1}^{k} \hat{P}(s, a_2, s_i)(r(s_i, a_1) - b(s_i, a_1))$$
$$= \frac{1}{2} - \frac{N(s, a_2, s_k)\tau}{N(s, a_2)} - \frac{(k+1)C_b\iota}{N(s, a_2)} - C_b\sqrt{\frac{\mathbf{Var}_{\hat{P}_{s,a_2}}(\mathbf{V})\iota}{N(s, a_2)}}.$$

With similar induction, we can prove that event $\{N(s, a_2) \geq \frac{Ck}{2\tau^2}\} \bigcap \{N(s, a_2, s_i) \in [\frac{C}{2\tau^2}, \frac{2C}{\tau^2}]\}_{i=1,\cdots,k}$ with probability over 0.8. When we have this concentration event true,

$$\mathbb{P}[\underline{\mathbf{Q}}(s, a_2) \leq \frac{1}{2} - \lambda\tau] \leq \mathbb{P}[-\frac{\tau}{4k} - C_b\tau^2\iota - \frac{C_b\tau^2\iota}{k} \leq -\lambda\tau].$$

By taking $\lambda = \frac{1}{4k} + C_b\tau\iota + \frac{C_b\tau^2}{k} + \epsilon$, where $\epsilon$ is a extremely small positive constant, $\mathbb{P}[\underline{\mathbf{Q}}(s, a_2) \leq \frac{1}{2} - \lambda\tau] = 0$. And further letting $\tau \leq \frac{1}{40\sqrt{\iota}}$, we have $\lambda \leq \frac{1}{4}C_b\sqrt{\iota}$. Putting the above inductions together,

$$\mathbb{P}[\xi_{bad}] \geq 1 - 0.1 - 0.2 - \exp(-\frac{1}{16}C_b^2\iota) \geq \frac{1}{2}.$$

This result points out that Algorithm 2 has a chance of over 1/2 to return a suboptimal policy. Therefore a overall coverage over all the optimal policies is necessary to derive a $\epsilon$-irrelevant bound for VI-LCB.