# OpenReview forum: "On Gap-dependent Bounds for Offline Reinforcement Learning"
_NeurIPS.cc/2022/Conference — NeurIPS 2022 Accept_

### Official Review · Reviewer_xbq6 · 2022-06-25

**Rating:** 7
**Confidence:** 3
**Soundness:** 4 excellent
**Presentation:** 3 good
**Contribution:** 3 good

**Summary:**

In this paper, the authors considers the offline MDP problem with an episodic setting. For this type of problem, existing works provide a minimax optimal sample complexity of  $O(H^3 S C^* \epsilon^{-2})$, given a finite single policy concentrability coefficient $C^*$. In this paper, the authors further extend the discussion on the problem based additional structure of a lower bounded Q-function gap or a lower bounded visitation probability of the behavioral policy on the support of the optimal policy. Based on these additional structures, $O(\frac{C^*}{\epsilon \cdot gap_{min}})$ or $O(\frac{1}{P \cdot gap_{min}^2})$ sample complexity are derived. Lower bounds are established and almost matches the provided upperbounds except for a factor of $H^2$.

Overall, the reviewer think this is a good submission that complete a missing part of the offline RL when the problem is simple (in terms of well lower bounded gaps). The discussion is complete and the presentation is clear.

**Questions:**

The reviewer may need to make some more explanation of what the subsampled VI-LCB (Algorithm 2) is doing. In particular, it will be helpful if the authors could add some intuitive discussion on why it is needed instead of simply applying the VI-LCB.

**Limitations:**

NA. This is a pure theoretical result.

**Strengths And Weaknesses:**

Strength #1. This paper complete a missing part of the offline RL when the problem has some strictly lower bounded gaps in optimal Q function or behavioral policy.

Strength #2. The discussion is complete in the sense that both lower and upper bounds are provided.

Strength #3. The presentation is clear and easy to follow.

Weakness: no obvious weakness found from the reviewer point of view.

---

> ### Author Response · Authors · 2022-08-02
> **Response to Reviewer xbq6**
>
>  Thank you for your appreciation! We'll keep modifying the details to make our work clearer and more complete in the final version.
> + The difference between Subsampled VI-LCB and naive VI-LCB lies in that Subsampled VI-LCB has an extra subsampling process of the dataset to wipe off the dependence in one trajectory. This technique reduces an order of $H$ in the sample complexity.
> + Actually, our analysis technique can also be applied to other LCB-style algorithms, including naive VI-LCB and PEVI-ADV [Xie, et al. 2021]. The reason why we choose Subsampled VI-LCB is that it's currently one of the optimal offline learning algorithms and relatively simpler to understand.

---

> ### Author Response · Authors · 2022-08-07
> **Any other questions to be addressed？**
>
> Thanks again for your review. We hope our answers could increase your confidence. As the discussion period is close to the end and we have not yet heard back from you, we would be glad to see if our rebuttal response has addressed your concerns questions/concerns.
> We are more than happy to discuss further if you have any further concerns and issues, please kindly let us know your feedback. Thank you for your time and help!

---

### Official Review · Reviewer_dSJR · 2022-07-08

**Rating:** 7
**Confidence:** 4
**Soundness:** 3 good
**Presentation:** 4 excellent
**Contribution:** 4 excellent

**Summary:**

This paper studies gap-dependent bounds for offline tabular RL. In particular, they show that under the optimal policy coverage assumption and the minimum positive gap, pessimistic algorithms achieve the rate of $1/N$. In the additional condition that the density of the behavior policy is uniformly lower bouned at the reachable states of an optimal policy, pessimistic algorithms precisely identify an optimal policy using finite samples. They also accompany with their upper bounds with nearly-matching lower bounds for each case. The paper also proposes the so-called deficit thresholding technique, based on the technique in an online counterpart, to analyze offline tabular RL with gap information.

**Questions:**

- Lemma 6.1 is particularly interesting as it gives a stronger upper bound on the sub-optimality when pessimism and gap information is used than when the gap information is ignored. However, I seem not to be able to locate the proof of Lemma 6.1 in the appendix. Do I miss something?
- Intuitively, what is the role of subsampling in Algorithm 2 and how does it contribute to the final bound?

**Ethics Review Area:**

["I don’t know"]

**Limitations:**

See the weakness section.

**Strengths And Weaknesses:**

## Strengths:
- the problem is novel and interesting
- the theoretical claims are sound and interesting to the offline RL community
- the analysis is quite general and neat, with a new technique called deficit thresholding that seems interesting
- every upper bound is backed up by nearly-matching lower bounds

## Weaknesses:
- The results and the techniques are only limited to tabular representation and require independence assumption on the episodes of the offline data.
- It is sometimes not trivial to locate the proof of the main result in the paper

---

> ### Author Response · Authors · 2022-08-02
> **Response to Reviewer dSJR**
>
> Thank you for your appreciation! We'll keep modifying the details to make our work clearer and more complete in the final version.
> + **Independence Assumption:** The independence assumption on the episodes of the offline data is widely adopted in previous papers [Rashidinejad et al., 2021, Yin and Wang, 2021, Xie and Jiang, 2021, Jin et al., 2021, Uehara and Sun, 2021, Uehara et al., 2021, Zanette et al., 2021]. On the other hand, it would be an interesting future direction to consider correlated datasets.
> + **Proof Location:** Sorry for the inconsistency of the same lemma in the appendix and the main paper. We decompose Lemma 6.1 into Definition A.5 and Theorem A.1 in the appendix. We have added explanations in the revised version.
> + **Subsampling:** Because the analysis of Subsampled VI-LCB is not the focus of this paper and the role of subsampling has been carefully discussed in Li et al. [2022], we just quote the lemmas without intuitive explanation. The role of subsampling is to wipe off the dependence between data points in one trajectory and avoid one extra dependence on H in the final complexity. We did not emphasize this because our analysis method can be applied to all LCB-style offline learning algorithms. Here Subsampled LCB is just chosen as one of the LCB-style algorithms with the best performance in minimax sample complexity.

---

### Official Review · Reviewer_xRSG · 2022-07-11

**Rating:** 5
**Confidence:** 5
**Soundness:** 3 good
**Presentation:** 3 good
**Contribution:** 3 good

**Summary:**

This paper provides an analysis of gap-dependent sample complexity in offline reinforcement learning. The main contribution is to show that under the optimal policy coverage assumption, the sample complexity can be improved to $O(1/\epsilon)$, compared to $O(1/\epsilon^2)$ bounds in previous analysis. Nearly matching lower bounds are also provided by the authors.

**Questions:**

This paper is well written and technically sound. I tend to vote for reject due to the following concerns and questions. Please address these in the rebuttal and correct me if I misunderstand some critical part of the work.

1. I am wondering why Theorem 4.1 cannot be implied by previous results by setting $\mathrm{gap}_{\min}=\epsilon$. If the output policy of an algorithm is $\epsilon$-optimal for some $\epsilon$ smaller than the gap, then we know that algorithm can identify the optimal policy. In fact, if I understand correctly, the proof of Theorem 4.1 indeed tries to argue this.

2. I like the analysis in general. By applying the clip technique, the authors can first upper bound the regret by considering a not-so-pessimistic-MDP, then applying existing techniques to continue the analysis. This makes the proof very clear and easy to follow for readers that are familiar with the literature. My main concern is that the main idea is from existing techniques in the literature. Although the paper argues that the novelty of the analysis is to consider the variance during clipping, this contribution seems to be a marginal contribution to me.

3. Theorem 5.1 shows the $\tilde{O}(1/\epsilon\mathrm{gap})$ upper bound. The authors argue that this significantly improves the sample complexity of $\epsilon$ is very small compared to the gap. I think this is a very nice contribution as it provides a sharper characterization of the sample complexity. However, I am wondering for $\epsilon>\mathrm{gap}$, does this also mean the proposed bound is actually worse than the existing results? Also, is $\epsilon << \mathrm{gap}$ really interesting? Again, if an algorithm can be $\mathrm{gap}$-optimal, we know it can identify the optimal policy.

4. I have some question related to the lower bound (Theorem 7.1). In the construction, C^* is fixed to be A. Does that mean this lower bound is only correct for some data collection policies?


5. One reason we prefer instance (or gap) dependent analysis over minimax analysis is that it can tell us if an algorithm is adaptively optimal. That is, if the algorithm is optimal for both hard and easy problem instances. However, as a recent paper pointed out, no algorithm can be instance-dependent optimal in offline RL (Xiao el al., 2021). This makes the contribution of this paper less interesting. Also, I think the paper should give some discussion about how the gap-dependent analysis considered in the paper is different with the instance-dependent analysis used in the literature (Lattimore and Szepesvari, 2020)


Xiao, C., Wu, Y., Mei, J., Dai, B., Lattimore, T., Li, L., Szepesvari, C. and Schuurmans, D., 2021, July. On the optimality of batch policy optimization algorithms. In International Conference on Machine Learning (pp. 11362-11371). PMLR.

Lattimore, T. and Szepesvári, C., 2020. Bandit algorithms. Cambridge University Press.



**Ethics Review Area:**

["I don’t know"]

**Limitations:**

Please see the previous section.

**Strengths And Weaknesses:**

Strength:

The paper is well-written and technically sound. Studying instance-dependent analysis for offline RL is a very important topic.


Weakness:

The proof technique is not very novel. The contributions might have limited impact.

---

> ### Author Response · Authors · 2022-08-02
> **Response to Reviewer xRSG**
>
> Thanks for your careful reading! We address **all of your concerns** below. We hope you may reevaluate your rating.
>
> Unless specified, the line numbers refer to the ones of the original submission.
>
> + **Setting $\epsilon = \mathrm{gap}_\min$ implies finding the optimal policy:** This is **NOT** correct.
>   - We first give a counterexample: Consider a contextual bandit with two states $s_1,s_2$ of same probability, and two actions $a_1, a_2$. Let the reward for $a_1,a_2$ be $(1,1-2\mathrm{gap}_\min)$ and $(1, 1-\mathrm{gap}_\min)$ respectively in two states. Then a $\mathrm{gap}\_{\min}$-optimal policy can be choosing $a_2$ in $s_1$ and $a_1$ in $s_2$. But such a policy is not an exact optimal policy.
>   - The key difference between minimax sample complexity and gap-dependent sample complexity in MDPs is that we need the suboptimality of every choice in the time-state pair $(h,s)$ reachable for any optimal policy is bounded by $\mathrm{gap}_\min$, while the previous result only cares about the suboptimality at the **initial state**. And this is also one of the main difficulties of generalizing bounds in bandits to bounds in MDPs.
>   - Although the statement is true for multi-armed bandits, simply setting $\epsilon < \mathrm{gap}_\min$ does not imply identifying the optimal policy in MDP as we have explained in line 160-161.  In addition, $\pi$ can also be suboptimal in the states visited at later timesteps.
> + **Clip Technique:** To the best of our knowledge, this is the first work that tries to translate the online gap-dependent analysis techniques to offline RL settings. The clipping with variance technique can be of great interest as it can be used to incorporate Bernstein bonus [Simchowitz and Jamieson, 2019]. We believe that this technique can potentially improve the dependency on $H$ in gap-dependent bounds for online MDPs.
> + **Significance of Upper Bounds:**
>   - All of our sample complexity upper bounds are for the same algorithm so it can directly adapt to different regions of $\epsilon$. Therefore, the true upper bound would be the minimum of all the bounds provided, including the previous $O(\epsilon^{-2})$ result.
>   - As we have explained in the first question, $\epsilon<\mathrm{gap}\_{\min}$ does not guarantee to identify an optimal policy. Therefore, $\epsilon<\mathrm{gap}\_{\min}$ is actually a non-trivial region.
> + **Lower Bounds (Theorem 7.1):** Thank you for your insightful comment on our lower bounds.
>   - We do have a stronger version of Theorem 7.1, which introduces a new variable to wipe off the relationship between $P$, $C^*$ and $A$, and this version of theorem is provided in the revised submission.
>   - On the other hand, the original version has already been strong enough. In line 124-129, we repeat the definition of offline learning, an instance of which is determined by a data collection policy (behavior policy) $\mu$ and an MDP $\mathcal{M}$ together. So gap-dependent minimax lower bound refers to the complexity with the worst given $(\mu,\mathcal{M})$ pair that has given $P,C^*,S,H,\mathrm{gap}\_{\min}$. Similar minimax lower bounds have been proposed in Dann et al. [2017], Yin et al. [2021a], Xie et al. [2021].
> + **Instance-dependent Optimal Algorithm:** "instance-optimal algorithm" (Xiao et al. [1]) is **different** from "gap-dependent optimal algorithm".
>   - "Instance-optimal" refers to the optimality for a **specific problem instance** in a reasonable algorithm family. In [1], they proved that in the minimax optimal algorithm family, no single algorithm can be optimal up to a constant factor in all the problem instances.
>   - "gap-dependent bound" in our paper refers to the **minimax optimality** over an **instance class** characterized by set a of parameters: $P,C^*,S,H,\mathrm{gap}\_{\min}$. Our results actually complement the hardness result in [1]. Although no reasonable algorithm can be optimal for all individual instances, we show that our algorithm is nearly optimal for each instance class specified  $(P,C^*,S,H,\mathrm{gap}\_{\min})$.
> + **Difference from Bandit Gap-Dependent Bounds**:
>   - As we have explained in the first point, in the MDP setting, $\epsilon<\mathrm{gap}\_{\min}$ does not imply optimality.
>   - As a result, clip technique is used for the RL setting to achieve the gap-dependent bounds.
> ```
> [1] Xiao, Chenjun, et al. "On the optimality of batch policy optimization algorithms." International Conference on Machine Learning. PMLR, 2021.
> [2] Lattimore, T. and Szepesvári, C., 2020. Bandit algorithms. Cambridge University Press.
> ```

---

> ### Author Response · Authors · 2022-08-07
> **Any other questions to be addressed？**
>
> Thanks again for your review. We hope our answers could increase your confidence. As the discussion period is close to the end and we have not yet heard back from you, we would be glad to see if our rebuttal response has addressed your concerns questions/concerns.
> We are more than happy to discuss further if you have any further concerns and issues, please kindly let us know your feedback. Thank you for your time and help!

---

> > ### Comment · Reviewer_xRSG · 2022-08-08
> > **Thanks for your response**
> >
> > Thanks for answering my questions. Most of my concerns have been addressed. I have adjusted my score accordingly :)
> >
> > **Setting $\epsilon=\mathrm{gap}_\min$ implies finding the optimal policy**
> >
> > Sorry that I was missing the definition of the optimal policy in Section 4. I think what confused me before is that in Section 3 it seems that the goal is to identify a near-optimal policy, but Section 4 actually considers identifying the optimal policy. Maybe it's helpful to include both objective in the problem setup?
> >
> > **Upper bounds and instance-dependent optimality**
> >
> > Thanks for the clarification. I think it would be helpful to include those discussions in the paper.
> >
> > **Lower bound**
> >
> > Thanks for providing a better bound.
> >
> > Sure I understand those minimax results consider the complexity given the worst $(\mu, \mathcal{M})$. But maybe a more interesting result would be that for any $\mu$, what's the sample complexity given the worst $\mathcal{M}$. Any comment on this?

---

> > > ### Author Response · Authors · 2022-08-08
> > > **Thanks for your reply**
> > >
> > > Thanks for your reply and for adjusting your score!
> > >
> > > + **Setting $\epsilon = \mathrm{gap}_\min$ implies finding the optimal policy:** Sorry about the confusion. We will emphasize the difference between identifying an optimal policy and a near-optimal policy in the final version.
> > > + **Significance of Upper Bounds:** Thanks for your suggestion! We will include these discussions in the final version.
> > > + **Lower Bounds:** Thanks for your suggestion! It is unclear if our lower bounds can be extended to any $\mu$ and the worst $\mathcal{M}$. In our current proof, we construct a specific $(\mu,\mathcal{M})$ pair to achieve the lower bound. If $\mu$ is given, then one probably needs to design a hard instance with a more sophisticated transition kernel and reward functions. We think this could be an interesting future direction.

---

### Official Review · Reviewer_P8NT · 2022-07-12

**Rating:** 6
**Confidence:** 4
**Soundness:** 3 good
**Presentation:** 3 good
**Contribution:** 3 good

**Summary:**

This paper studies the gap dependent sample complexity of obtaining the optimal policy in episodic RL with offline data. The authors first define the notions of uniform optimal policy coverage coefficient and the relative optimal policy coverage. The uniform optimal policy coverage characterizes how "explorative" is the offline sampling policy, and relative optimal policy coverage characterizes how different are the offline sampling policy and the optimal policy. Further, the authors establish an upper bound on the sample complexity of offline RL using the optimal coverage coefficient. Next, the authors argue that optimal coverage coefficient might be too small. Hence, they provide a sample complexity based on relative optimal policy coverage coefficient. Next, the authors provide the main proof techniques in their results. Finally, the authors provide a gap dependent lower bound on the sample complexity of their algorithm.

**Questions:**

- In Assumption 3.1, are you implicitly assuming only deterministic optimal policies? The reason is that you are denoting \pi^*(s), which is the action taken at state s in the deterministic policy \pi^*.
- In the result of Theorem 4.1, why do you hide the dependency on log(1/\delta)? I believe it is more informative if you somehow represent that dependency here.
- In Theorem 7.1 you introduce the notion of ALG, but you have not defined this notion.
- The lower bound on Theorem 7.1 and Corollaries 7.1 and 7.2 are in terms of expectation. But the upper bound in Theorems 5.1 and 5.2 are in high probability. How do you relate these two?

**Limitations:**

As mentioned earlier, the upper bound in the paper is stated in terms of high probability, but the lower bound is in terms of expectation. If the authors address this concern of mine, I am willing to increase my score.

**Strengths And Weaknesses:**

The main strength of the paper is a through study of offline RL in the episodic case. In particular, the authors provide both an upper bound and a lower bound, and they claim that they match up to some polynomial of length of the horizon H.
The main weakness is on the appearance of gap_m in the sample complexity. Although I understand this is the main motivation of the paper from the beginning, but this quantity can be arbitrary small. The authors might claim that gap_m also appears on the lower bound, and hence their bound is tight. But as I explained before, I am not yet convinced by the lower bound, since it is in terms of expectation rather than high probability.
Another issue is on the definition of ALG in the lower bound. This notion is not defined anywhere in the paper. This is critical, since it tells us about the kind of lower bound that the authors are claiming.

---

> ### Author Response · Authors · 2022-08-02
> **Response to Reviewer P8NT**
>
> Thanks for your appreciation. We address your questions below.
> + **Assumption 3.1:** Sorry for the confusion in Assumption 3.1. $\pi^*$ is an arbitrary (possibly stochastic) optimal policy. Note that states that are reachable by stochastic optimal policies are also reachable by deterministic optimal policies, so constraining $\pi^*$ to deterministic optimal policy works as well. We have fixed the expression.
> + **Dependence on $\log(1/\delta)$:** The log terms in Theorem 4.1, Theorem 5.1 and Theorem 5.2 were all hidden in the $\widetilde{O}$ for simplicity. The log terms are of power 1 in all three theorems. Full analysis without hiding the log term is in Appendix B.3, B.4, B.5. We have included the $\delta$ term in the revised version.
> + **Missing definition of ALG:** Thank you for pointing out this! ALG is defined in line 684 and line 685 in the appendix (original version). We have put it in the main paper (line 251 to line 253 in the revised version).
> + **Expected Lower Bound:**
>   - First, it is common to provide expected lower bounds, e.g., in Xie et al [2021] and Rashidinejad et al [2021].
>   - Second, an expected upper bound follows directly from the PAC upper bound. Simple calculations show that the maximum expectation of suboptimality is $\epsilon +\delta H \mathrm{gap}\_\mathrm{min}$. Taking $\delta=\frac{\epsilon}{H\mathrm{gap}_{\mathrm{min}}}$ makes it a $O(\epsilon)$ suboptimality.
>   - Third, the hard instance we constructed also permits the same lower bound with constant probability, which is another commonly used lower bound. The intuition is that in the constructed MDP family (see C.2.1 in the appendix for details), the suboptimality of any policy is upper bounded by $\lambda H\tau$, so $\hat{\pi}$ must suffer from a $\Omega(\lambda H\tau)$ suboptimality under some constant probability when the expectation of suboptimality is $\Omega(\lambda H\tau)$.
>   - Thank you for reminding us of the inconsistency in upper bounds and lower bounds. We have added a brief proof for the constant-probability lower bound in the revised version (Appendix C.2.3).

---

> ### Author Response · Authors · 2022-08-07
> **Any other questions to be addressed？**
>
> Thanks again for your review. We hope our answers could increase your confidence. As the discussion period is close to the end and we have not yet heard back from you, we would be glad to see if our rebuttal response has addressed your concerns questions/concerns.
> We are more than happy to discuss further if you have any further concerns and issues, please kindly let us know your feedback. Thank you for your time and help!

---

### Meta-Review · Area_Chair_mKw7 · 2022-08-22

**Recommendation:** Accept
**Confidence:** Certain

**Metareview:**

This paper studies gap-dependent sample complexity in offline tabular RL. The authors show that when there is a gap in the optimal Q-function (and the density ratio between optimal and behavior policies is upper-bounded), the sample complexity can be improved from $O(1/\epsilon^2)$ to $O(1/\epsilon)$ using a pessimistic algorithm. The authors also provide gap-dependent lower bounds.

The work seems correct and well-executed. It is of somewhat limited impact given the strong assumptions (tabular MDP, coverage) but fills a gap (pun intended) in the literature and the results are interesting enough to warrant acceptance.


**Award:**

No

---

### Decision · Program_Chairs · 2022-09-14

Accept